# Plastic roles of pericytes in the blood–retinal barrier

Do Young Park[1], Junyeop Lee[1], Jaeryung Kim[1], Kangsan Kim[2], Seonpyo Hong[1], Sangyeul Han[2], Yoshiaki Kubota[3], Hellmut G. Augustin[4], Lei Ding[5], Jin Woo Kim[6], Hail Kim[1], Yulong He[7], Ralf H. Adams[8] & Gou Young Koh[1,2]

The blood–retinal barrier (BRB) consists of tightly interconnected capillary endothelial cells covered with pericytes and glia, but the role of the pericytes in BRB regulation is not fully understood. Here, we show that platelet-derived growth factor (PDGF)-B/PDGF receptor beta (PDGFRβ) signalling is critical in formation and maturation of BRB through active recruitment of pericytes onto growing retinal vessels. Impaired pericyte recruitment to the vessels shows multiple vascular hallmarks of diabetic retinopathy (DR) due to BRB disruption. However, PDGF-B/PDGFRβ signalling is expendable for maintaining BRB integrity in adult mice. Although selective pericyte loss in stable adult retinal vessels surprisingly does not cause BRB disintegration, it sensitizes retinal vascular endothelial cells (ECs) to VEGF-A, leading to upregulation of angiopoietin-2 (Ang2) in ECs through FOXO1 activation and triggering a positive feedback that resembles the pathogenesis of DR. Accordingly, either blocking Ang2 or activating Tie2 greatly attenuates BRB breakdown, suggesting potential therapeutic approaches to reduce retinal damages upon DR progression.

[1] Graduate School of Medical Science and Engineering, Korea Advanced Institute of Science and Technology (KAIST), Daejeon 34141, Korea. [2] Center for Vascular Research, Institute of Basic Science (IBS), Daejeon 34141, Korea. [3] The Laboratory of Vascular Biology, Keio University, Tokyo 160-8582, Japan. [4] Division of Vascular Oncology and Metastasis, German Cancer Research Center, DKFZ-ZMBH Alliance, 69120 Heidelberg, Germany. [5] Columbia Stem Cell Initiative, Department of Rehabilitation and Regenerative Medicine, Department of Microbiology and Immunology, Columbia University Medical Center, New York, New York 10032, USA. [6] Department of Biological Sciences, KAIST, Daejeon 34141, Korea. [7] Cyrus Tang Hematology Center, Collaborative Innovation Center of Hematology, Soochow University, Suzhou 215123, China. [8] Department of Tissue Morphogenesis, Max-Planck-Institute for Molecular Biomedicine, and Faculty of Medicine, University of Münster, D-48149 Münster, Germany. Correspondence and requests for materials should be addressed to G.Y.K. (email: gykoh@kaist.ac.kr).

The blood–retinal barrier (BRB) is divided into inner and outer parts[1,2]. The inner part consists of tightly connected capillary endothelial cells (ECs) covered with pericytes and Müller glial cells, whose function is to nourish the inner two thirds of the retina. The outer part of the BRB consists of tightly connected pigment epithelial cells that maintain the integrity of the outer third of the retina. Pericytes differ in origin, morphology, and function depending on different organs' vascular beds and are located between ECs and surrounding parenchymal cells, sharing a basement membrane with the endothelium[3–6]. They play important roles in a range of functions, including angiogenesis, vascular remodelling, regression and stabilization, and generation and maintenance of the blood–brain barrier (BBB) and BRB[5–7]. Previous studies[8–14] revealed that proper pericyte recruitment to growing retinal and brain vessels through controlled platelet-derived growth factor (PDGF)-B/PDGF receptor beta (PDGFRβ) signalling is critical for adequate formation of the BRB and BBB. Genetic deletion of PDGF-B in ECs or its retention motif that is responsible for the retention of PDGF-B within the pericellular space in the growing retinal vessels exhibited severe vascular impairments including vascular engorgement and leakage together with improper pericyte recruitment and organization during the retinal vascular development[9,12], which are similar to the events induced by a PDGFRβ blocking antibody[11]. These studies indicated that PDGF-B secreted from and properly retained in retinal vascular ECs recruits PDGFRβ[+] pericytes to facilitate vascular growth and stabilization during vascular development. However, whether PDGF-B/PDGFRβ signalling is continuously required for adequate pericyte attachment to stable adult retinal vessels remains unclear. Moreover, why adequate pericyte attachment is critical for maintenance of BRB integrity in adult retinal vessels is unknown.

Pericyte dropout or loss is one of the hallmarks of diabetic retinopathy (DR) and it has been postulated to initiate or trigger several pathologic features including microaneurysm formation, abnormal leakage, edema and ischemia, provoking proliferative neovascularization in the retina[15–17]. Disturbance of the angiopoietin–Tie2 system is closely linked with DR pathogenesis[18]. Pericyte-derived angiopoietin-1 (Ang1) is known to stimulate Tie2 signalling in ECs, which may be substantial contributing factor for a stable interaction between ECs and pericytes in adult[19–22]. However, a concrete evidence to support this concept is not yet available. In contrast, Ang2 secreted from activated or inflamed ECs destabilizes EC–pericyte interaction by inhibiting Ang1–Tie2 signalling as a partial antagonist[19]. In fact, increased Ang2 in pathologic conditions, including DR, is thought to be one of the major factors causing pericyte dropout, and antagonizing Ang1–Tie2 signalling[23]. However, endogenous resources and the exact roles of Ang1, Ang2 and Tie2 in building up BRB during development and in the disintegrating BRB during pathologic conditions need to be better defined.

In this study, we show that PDGF-B/PDGFRβ signalling, which is critical for BRB buildup in growing retinal vessels, is dispensable for BRB integrity maintenance in adult mice. Selective pericyte loss in adult stable retinal vessels surprisingly did not result in phenotypes of BRB disintegration. The vessels did, however, show susceptibility to leakage upon external VEGF-A stimulus, which could be mediated by the elevated vascular destabilizing factors including Ang2 and VEGFR2 through transcriptional activation of FOXO1 in unstable endothelial cells. This process was inhibited by Ang2 blockade or Tie2 activation, reducing retinal damages during DR progression by inhibiting BRB disintegration.

## Results

**Pericyte coverage is critical for BRB maturation.** To gain an insight into the role of PDGF-B/PDGFRβ signalling in BRB formation, we first examined their expression in growing retinal vessels at postnatal day 5 (P5) in the normal C57BL/J mouse. PDGF-B was abundantly and selectively expressed in ECs with the highest expression in the vascular front region, while PDGFRβ was selectively expressed in pericytes that covered most ECs except the tip ECs (Fig. 1a,b), consistent with previous findings[12]. Next, to deplete PDGF-B in retinal vessels in tamoxifen-inducible manner, we crossed the *Pdgfb*[flox/flox] mouse[9] with the *VE-cadherin*-Cre-ER[T2] mouse[24] and produced a *Pdgfb*[iΔEC] mouse (Fig. 1c). Cre-ER[T2] positive but flox/flox negative littermates were defined as wild-type (WT) mice in each experiment. When PDGF-B was specifically depleted in ECs of *Pdgfb*[iΔEC] mice from P5, at P12 the mice had severe hemorrhage at both the inner and outer surfaces of the retinal cup, profound accumulation of whitish inflammatory cells, retinal detachment (~50%), enlarged but well-perfused vessels, severely impaired pericyte coverage of the vessels, microaneurysm formation, reduced ZO-1 and disarranged VE-cadherin, red blood cell (RBC) leakage, profound macrophage infiltration, little or no deep vascular plexus formation, extreme hypoxia with increased VEGF-A, and upregulation of the genes (CCL-2, CCL-3, TNF-α and VEGF-A) involved in inflammation and hypoxia in retinas compared with those of WT mice (Fig. 1c–h; Supplementary Fig. 1). These results indicate that EC-derived PDGF-B plays indispensable roles in vascular remodelling and BRB maturation through active recruitment of pericytes onto growing retinal vessels, and that impairment of PDGF-B/PDGFRβ signalling during this period exhibits multiple hallmarks of DR.

**Inadequate pericyte coverage impairs BRB and vision.** These findings led us to ask whether the impaired vascular features can be normalized or sustained as the mice mature. To answer this question, PDGF-B was depleted from P5 in the *Pdgfb*[iΔEC] mice, and the retinas were examined at P30 (Fig. 2a). Severe hemorrhage and RBC leakage, disrupted vascular plexus formations in all three layers with poor pericyte coverage, profound macrophage infiltration, detachment of smooth muscle actin[+] (SMA[+]) cells from retinal artery, highly increased EC apoptosis, and extreme hypoxia were still detected in the retina together with impaired vision in the *Pdgfb*[iΔEC] mice (Fig. 2b–h). These data indicate that impaired BRB function caused by the initial poor pericyte coverage is permanent and beyond self-repair, and could continuously worsen neuronal function in the retina, eventually leading to blindness.

We hypothesized that severe ischemia due to the impaired vascular features might be a major driving factor in this irreversible change. To address this hypothesis, we generated an oxygen-induced retinopathy (OIR) model in the *Pdgfb*[iΔEC] mice (Supplementary Fig. 2a). OIR caused more severe hemorrhage, RBC leakage, hypoxia and macrophage infiltration together with a larger avascular area but smaller neovascular tuft area in *Pdgfb*[iΔEC] mice compared with WT mice (Supplementary Fig. 2b–f). Thus, adequate pericyte coverage is required to prevent BRB impairment against further vascular impairments, including severe ischemia, inflammation and oxidative stresses.

**Inadequate pericyte upregulates FOXO1 and Ang2 in ECs.** We then asked how inadequate pericyte coverage affects the impairments of retinal vessels and BRB. We focused on analysing the expression and roles of Ang2 and Tie2, given that these are critical factors regulating vascular stabilization and integrity. Consistent with previous reports[25–27], Ang2 was high in the

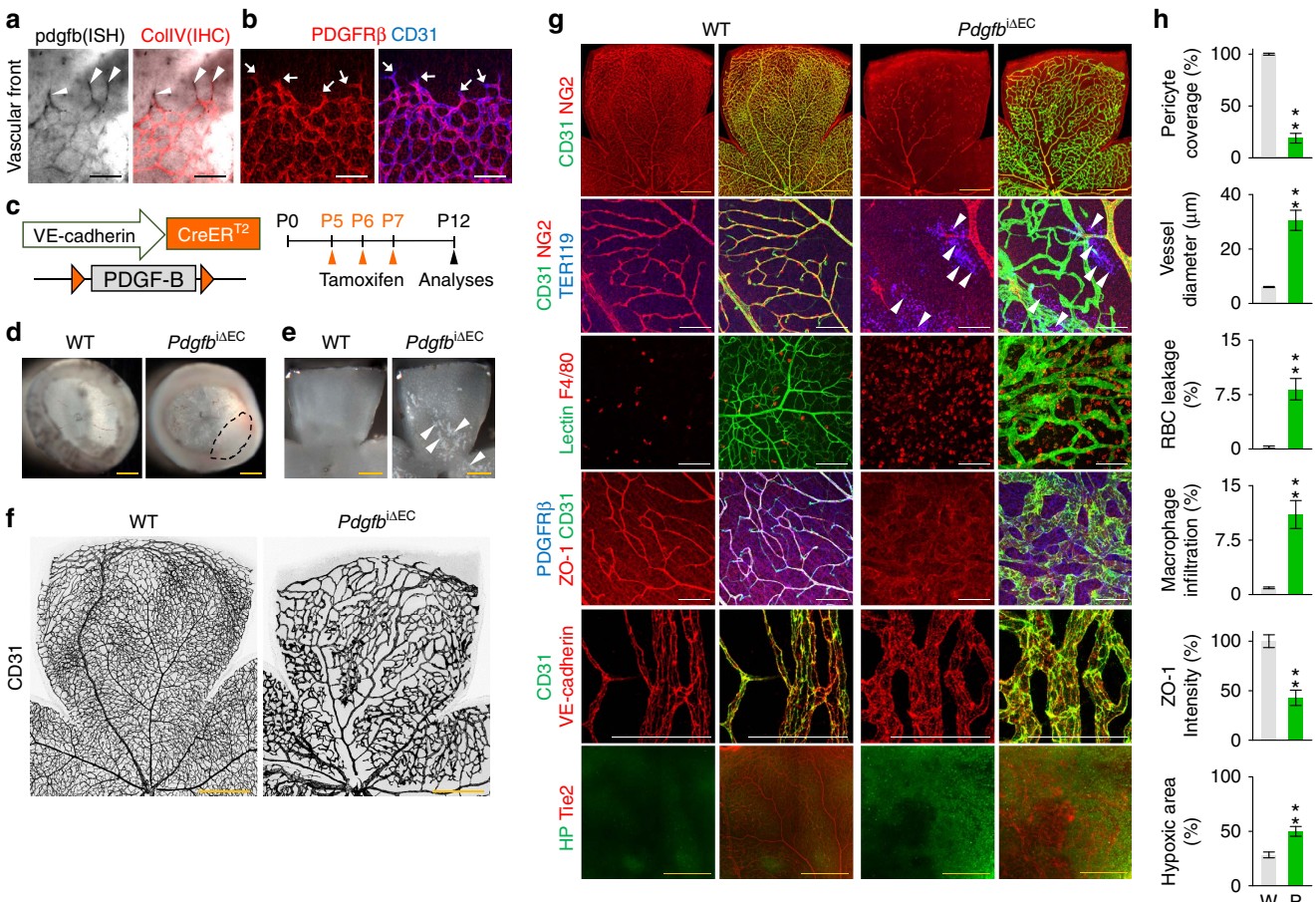

**Figure 1 | Pericyte coverage is critical for formation and maturation of BRB.** (**a**) Abundant PDGF-B mRNA expression in the tip ECs of vascular front (white arrowheads) at P5. PDGF-B mRNA was analysed by *in situ* hybridization (ISH), and then collagen type IV (ColIV) was detected by immunohistochemistry (IHC). (**b**) Immunostaining for PDGFRβ+ pericytes and CD31+ ECs in vascular front at P5. Note that there is no pericyte coverage in tip ECs (arrows). (**c**) Diagram for EC-specific depletion of PDGF-B in retinal vessels from P5 and their analyses at P12 using *Pdgfb*^iΔEC mice. (**d**–**f**) Images of hemorrhage (red) and retinal detachment (dashed-line circle) in inner surface of retinal cup, and whitish inflammatory cell accumulations (white arrowheads) and irregular CD31+ vascular plexus in retinas of *Pdgfb*^iΔEC mice are shown. (**g**) Images of NG2+ and PDGFRβ+ pericyte coverage onto CD31+ vessels, TER119+ RBC leakage (white arrowheads), F4/80+ macrophage infiltration, lectin perfusion, distributions of ZO-1 and VE-cadherin onto CD31+ ECs, and hypoxic area in retinas of WT and *Pdgfb*^iΔEC mice are shown. HP, hypoxyprobe. (**h**) Comparisons of indicated parameters in WT (W; n = 6) and *Pdgfb*^iΔEC (P; n = 6) mice. Error bars represent mean ± s.d. **P < 0.01 versus WT by Mann–Whitney *U* test. Scale bars, 100 μm (white, black) and 500 μm (yellow).

pericyte-free tip ECs but low in ECs of the phalanx and plexus, while Tie2 expression showed the opposite pattern in the normally growing retinal vessels (Fig. 3a,b). This finding led us to hypothesize that pericyte coverage *per se* might regulate the expression of Ang2 or Tie2 in the adjacent ECs. Intriguingly, the inadequate pericyte attachment markedly increased Ang2 but reduced Tie2 in the ECs of retinal vessels (Fig. 3c–e; Supplementary Fig. 3a,b), which is similar to the findings in advanced cancers[28]. Moreover, the ECs of micro-aneurysms in the pericyte-detached vessels of *Pdgfb*^iΔEC mice or those of OIR mice had relatively higher Ang2 (Fig. 3d; Supplementary Fig. 4a,b).

Because Ang2 is upregulated together with ESM1 by the enhanced transcriptional activity of forkhead box protein O1 (FOXO1)[29–31], we examined the changes of these molecules *in vivo* and *in vitro*. Similar upregulation in the expressions of ESM1 and VEGFR2, which are highly enriched in the pericyte-uncovered tip ECs[25,32], and distinct nuclear localization of FOXO1 were found in the ECs of pericyte-uncovered retinal vessels (Fig. 3f–i; Supplementary Fig. 3c). In addition, mRNA levels of Ang2 and ESM1 were increased upon Tie2 silencing in primary cultured HUVECs, and these changes were FOXO1 dependent (Supplementary Fig. 4c). Furthermore, Tie1 was

reduced in the ECs of pericyte-detached retinal vessels, while soluble Tie1 (cleaved ecto-domain) was increased in the vitreous fluid of *Pdgfb*^iΔEC mice (Fig. 3h,i); these are similar to those which occur to ECs upon acute inflammation[33,34]. These data suggested that the ECs became unstable due to inadequate pericyte coverage, and that cleavage of Tie1 led to the suppression of Tie2 availability, which is required to inhibit Ang2 transcription by FOXO1 (refs 33,34).

Next, to reveal whether and how pericyte coverage regulates FOXO1 and Ang2, we employed an *ex vivo* aortic ring assay[35] using the aortas of WT and *Pdgfb*^iΔEC mice, where the mice were pre-administrated with tamoxifen before aorta harvest (Supplementary Fig. 5a). During *ex vivo* culture, the PDGFRβ+ pericytes were detached from the sprouts, which were reduced in number and length themselves, of the aorta of *Pdgfb*^iΔEC mice, and the pericyte-detached ECs had higher levels of Ang2 and FOXO1 as well as nuclear localization of FOXO1 (Supplementary Fig. 5b–e). In contrast, no major changes in FOXO1 or Ang2 were detected in the ECs of sprouts with intact pericytes in the aorta of WT mice (Supplementary Fig. 5d,e). Thus, pericyte detachment appears to directly upregulate and activate FOXO1 in ECs, which leads to stimulation of Ang2 expression in ECs. In

fact, Chip-PCR sequencing analysis revealed that FOXO1 binding was enriched in the promoter and enhancer regions of Ang2 (Supplementary Fig. 6), indicating that FOXO1 directly regulates Ang2 expression in the destabilized ECs caused by inadequate pericyte coverage.

**Pericytes of retinal vessels are not the main source of Ang1.** Because pericyte-derived Ang1 contributes to vascular stabilization[19–22], we investigated the expression and role of Ang1 using *Ang1*-GFP knock-in[36] and *Ang1*$^{flox/flox}$ (ref. 37) mice in the growing retinal vessels. Unexpectedly, however, pericytes

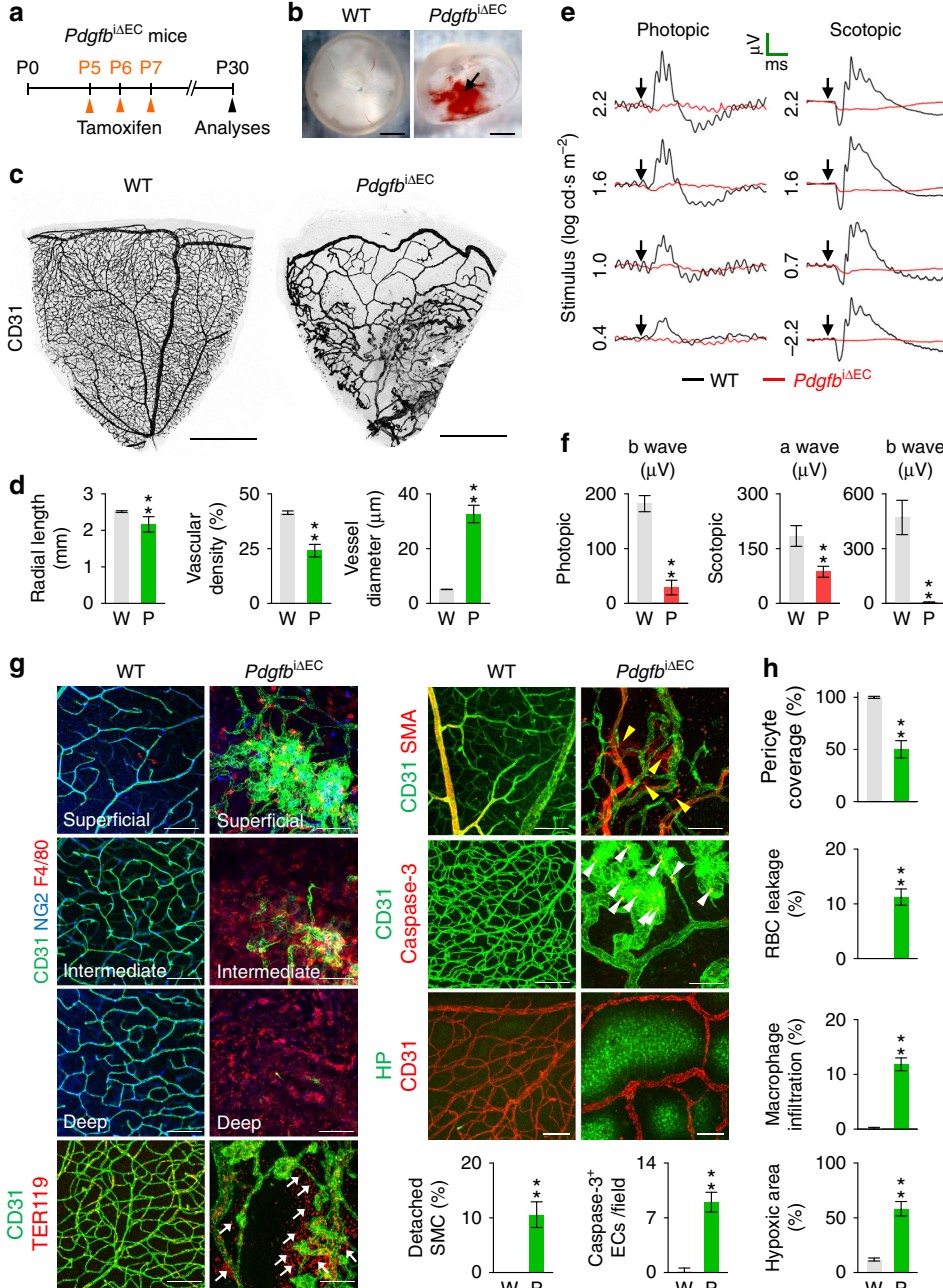

**Figure 2 | Inadequate pericyte coverage impairs BRB and vision.** (**a**) Diagram depicting the experiment schedule for EC specific deletion of PDGF-B in retinal vessels from P5 and their analyses at P30 in *Pdgfb*$^{iΔEC}$ mice. (**b,c**) Images of inner surface and vascular plexus in retinas of WT and *Pdgfb*$^{iΔEC}$ mice. Vitreous hemorrhage (arrow) and severely impaired vascular network are detected in *Pdgfb*$^{iΔEC}$ mice. (**d**) Comparisons of indicated parameters in WT (W; $n = 6$) and *Pdgfb*$^{iΔEC}$ (P; $n = 6$) mice. (**e**) Electroretinography in WT and *Pdgfb*$^{iΔEC}$ mice. Normal ERG responses showing photopic b wave and scotopic a and b waves in WT mice (black line) are hardly detected in *Pdgfb*$^{iΔEC}$ mice (red line). Arrows, flash stimuli. Scale bars (green), 40 ms ($x$) and 40 μm ($y$) in photopic ERG, 40 ms ($x$) and 160 μm ($y$) in scotopic ERG. (**f**) Comparisons of amplitude of photopic b wave and scotopic a and b waves by the stimulus of 1.6 log cd s m$^{-2}$ in WT (W; $n = 6$) and *Pdgfb*$^{iΔEC}$ (P; $n = 6$) mice. (**g**) Images showing impaired CD31$^+$ vascular plexus in superficial, intermediate and deep layers, enlarged CD31$^+$ vessels, detached or no NG2$^+$ pericyte coverage, abundant F4/80$^+$ macrophage infiltration and TER119$^+$ RBC leakage (arrows), severe hypoxia, detached SMA$^+$ SMCs from the CD31$^+$ vessels (yellow arrowheads), and caspase-3$^+$ apoptotic ECs (white arrowheads) are largely detected in retinas of *Pdgfb*$^{iΔEC}$ mice compared with those of WT mice. HP, hypoxyprobe. (**h**) Comparisons of indicated parameters in WT (W; $n = 6$) and *Pdgfb*$^{iΔEC}$ (P; $n = 6$) mice. Error bars represent mean ± s.d.**$P < 0.01$ versus WT by Mann–Whitney $U$ test. Scale bars, 100 μm (white) and 500 μm (black).

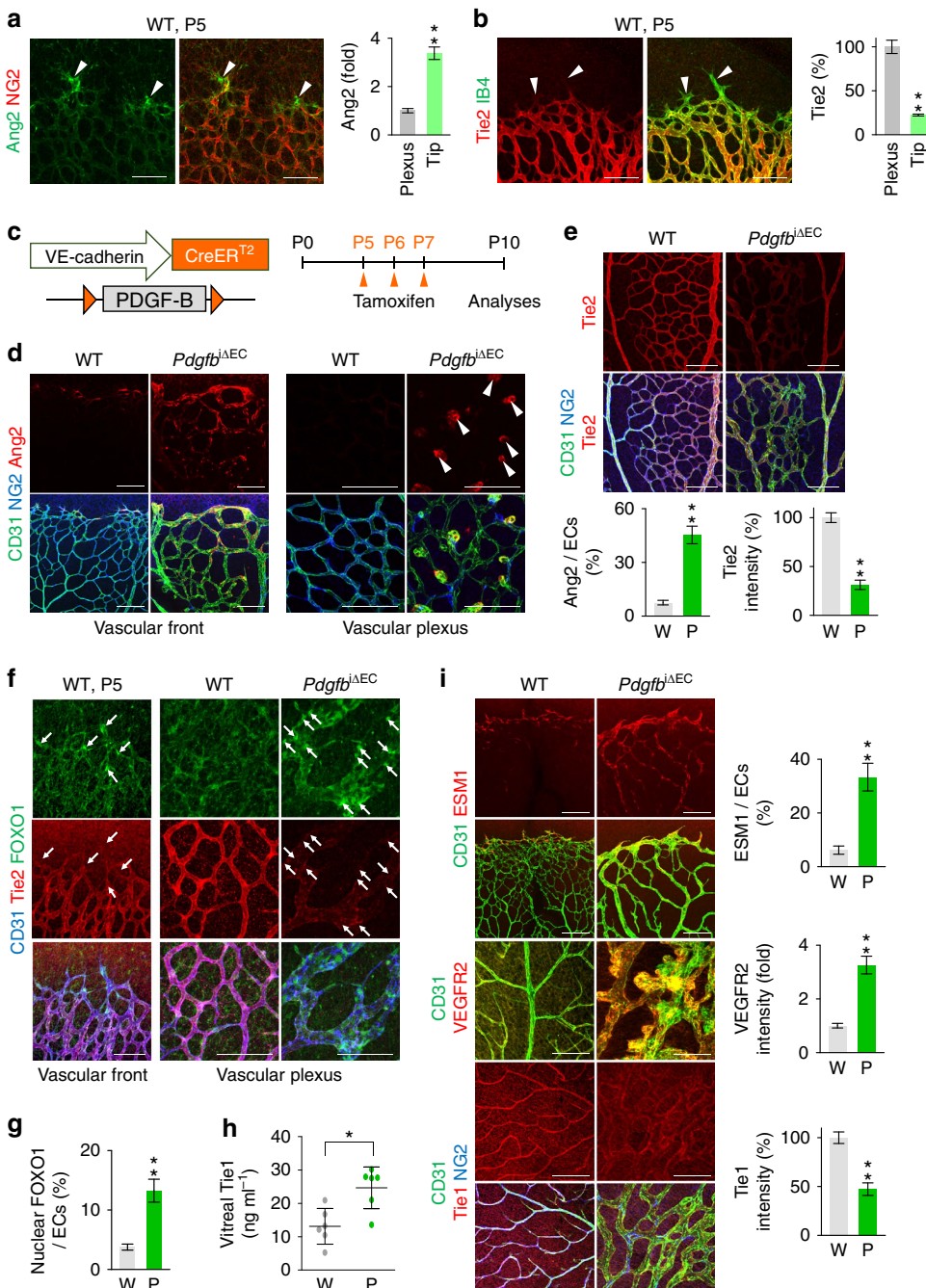

**Figure 3 | Inadequate pericyte coverage activates FOXO1–Ang2 axis in ECs.** (**a,b**) Images and comparisons of Ang2 and Tie2 expressions in vascular front of retinas of WT mice at P5. Tip ECs that are not covered with NG2+ pericytes (arrowheads) express high Ang2 and low Tie2. Analyses were performed in 6 representative areas per retina in WT mice ($n = 6$). (**c**) Diagram depicting the experiment schedule for EC-specific deletion of PDGF-B in retinal vessels from P5 and their analysis at P10 in $Pdgfb^{i\Delta EC}$ mice. (**d,e**) Images and comparisons of Ang2 and Tie2 expressions in CD31+ vessels. Increased expression of Ang2 are observed in vascular front and microaneurysm, while Tie2 expression is decreased in retinal vessels of $Pdgfb^{i\Delta EC}$ mice (P; $n = 6$) compared with WT mice (W; $n = 6$). (**f**) Expressions and localizations of FOXO1 and Tie2 in vascular front of WT mice at P5 and in vascular plexus at P10 in WT mice and $Pdgfb^{i\Delta EC}$ mice. Note that nuclear localization of FOXO1 (arrows) is evident in Tie2-low tip ECs of WT mice and pericyte-detached ECs in $Pdgfb^{i\Delta EC}$ mice. (**g**) Quantification of the nuclear FOXO1 in the CD31+ vessels in WT (W; $n = 6$) and $Pdgfb^{i\Delta EC}$ (P; $n = 6$) mice. (**h**) Comparison of soluble Tie1 in vitreous fluid of eyes of WT (W; $n = 6$) and $Pdgfb^{i\Delta EC}$ (P; $n = 6$). (**i**) Expressions, localizations, and comparisons of ESM1, VEGFR2 and Tie1 in CD31+ retinal vessels of WT (W; $n = 6$) and $Pdgfb^{i\Delta EC}$ (P; $n = 6$). Error bars represent mean ± s.d. *$P < 0.05$, **$P < 0.01$ versus WT by Mann–Whitney $U$ test. All scale bars, 100 μm.

covering retinal vessels did not express Ang1, but those along choroidal vessels and neuronal cells in the ganglion cell and inner nuclear layers of the retina did (Fig. 4a–c). To confirm this unexpected finding, we generated $Ang1^{i\Delta PC}$ mice by crossing

$Ang1^{flox/flox}$ mice[37] with $PDGFR\beta$-$Cre$-$ER^{T2}$ mice[38] (Supplementary Fig. 7a). High efficiency was shown for targeting PDGFRβ+ pericytes in retinal vessels by tamoxifen-induced Cre-mediated recombination using the $PDGFR\beta$-$Cre$-$ER^{T2}$ mice

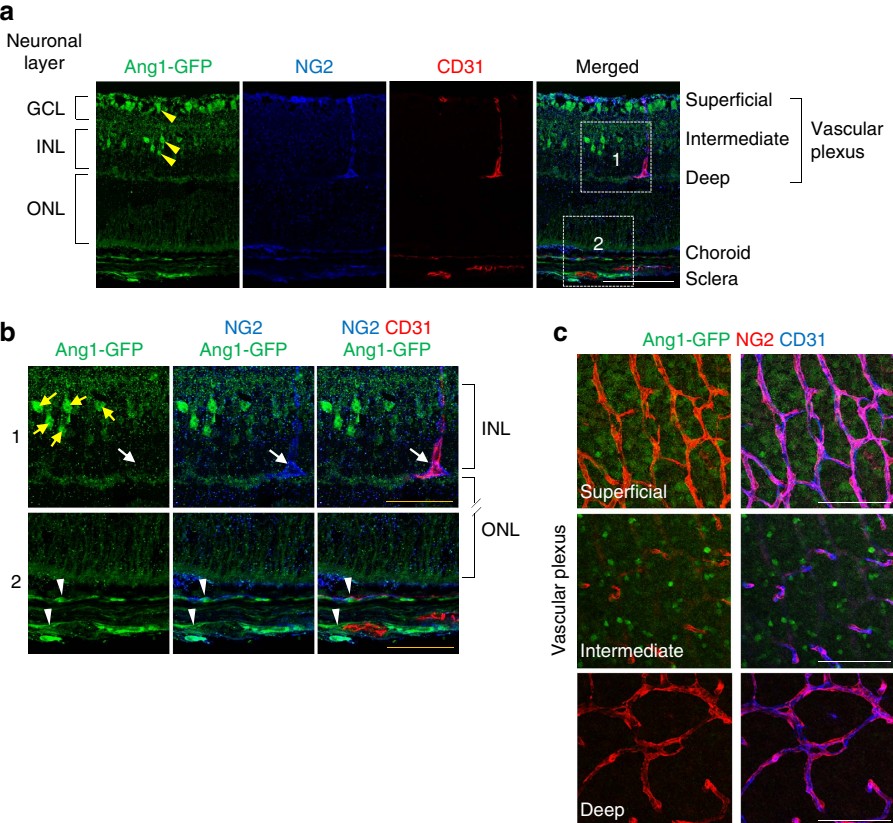

**Figure 4 | Pericyte is not a source of Ang1 in the retinal vessels. (a,b)** Cross-sectional images of retina of *Ang1*-GFP knock-in mouse at P5. Note that most of GFP signals are detected in the retinal neurons (yellow arrowheads) located in ganglion cell layer and inner nuclear layer. Magnified images of boxed areas are shown below. No GFP signal was detected in NG2$^+$ pericytes around retinal vessels (arrows), whereas positive GFP signal was detected in NG2$^+$ pericytes around choroidal vessels (white arrowheads). **(c)** Images showing localization of GFP$^+$ cells in relation to CD31$^+$ retinal vessels and NG2$^+$ pericytes in superficial, intermediate, and deep retinal vascular layers. GFP signals are not detected in pericytes or ECs, but detected in neurons that surround retinal vessels in superficial and intermediate vascular plexus layers. Scale bars, 100 μm (white) and 50 μm (yellow).

(Supplementary Fig. 8). Selective depletion of Ang1 in PDGFRβ$^+$ pericytes from birth did not yield any apparent differences in gross features, growth, network, EC proliferation, RBC localization, pericyte coverage, or expression of Tie2 and Ang2 in the retinal vessels at P5 compared with those of WT mice (Supplementary Fig. 7b–d). These results clearly indicate that retinal pericytes do not produce Ang1 to regulate vascular growth and BRB maturation.

**Tie2 is critical for retinal vessel growth and stabilization.** In comparison, when Tie2 was specifically depleted in the ECs by crossing *Tie2$^{flox/flox}$* mice[27] with *VE-Cadherin-Cre-ER$^{T2}$* mice (*Tie2$^{iΔEC}$* mice) (Fig. 5a), the retinal vessels showed moderate hemorrhage in the inner surface of the retinal cup, severely disrupted vascular network formation, enlarged diameter but short radial length, less density and branching, lack of deep vascular plexus, and profound RBC leakage and macrophage infiltration without a significant difference in pericyte coverage at P15 compared with those of WT mice (Fig. 5b–g). No normalization was observed until P30 in the retinas of *Tie2$^{iΔEC}$* mice, which showed vascular leakage with engorged and disorganized vessels (Supplementary Fig. 9). We also noted stronger nuclear localization of FOXO1 and expression of Ang2 in the Tie2-depleted ECs of retinal vessels (Fig. 5f,g). However, when Tie2 was specifically depleted in the ECs at adulthood, no noticeable changes including vascular leakage were detected in

the retinal vessels compared with those of WT mice (Supplementary Fig. 10). These results indicate that Tie2 is indispensable for vascular growth and BRB maturation in the growing retinal vessels, but is dispensable for the maintenance of BRB integrity during adulthood. In fact, lung vessels continue to require Tie2 (ref. 39), while the retinal vessels no longer require Tie2 during adulthood, suggesting that dependency on Tie2 for vascular stability and integration could be organ-specific during adulthood.

**Inactivation of FOXO1 or Ang2 restores vascular impairments.** To address how FOXO1-induced Ang2 upregulation contributes to vascular impairments in pericyte-uncovered retinal vessels, we crossed the *Foxo1$^{flox/flox}$* mouse[40] with the *VE-cadherin-Cre-ER$^{T2}$* mouse and generated the *Foxo1$^{iΔEC}$* mouse (Fig. 6a). Then, inadequate pericyte coverage in the *Foxo1$^{iΔEC}$* mouse was generated by intra-peritoneal administration of the PDGFRβ$^+$ blocking antibody APB5 (ref. 11) at P1 (Fig. 6a). *Foxo1$^{iΔEC}$* mice without APB5 treatment showed a mild impairment of vascular branching in the distal part of the retina when FOXO1 was depleted from P5 (Fig. 6b–e), but moderate impairment in sprouting angiogenesis when it was depleted from P1 (Supplementary Fig. 11). On the other hand, the findings of APB5 treatment in the retinal vessels of WT mice faithfully recapitulated those of *Pdgfb$^{iΔEC}$* mice, such as severe hemorrhage at the inner surface of the retinal cup, impaired retinal vascular

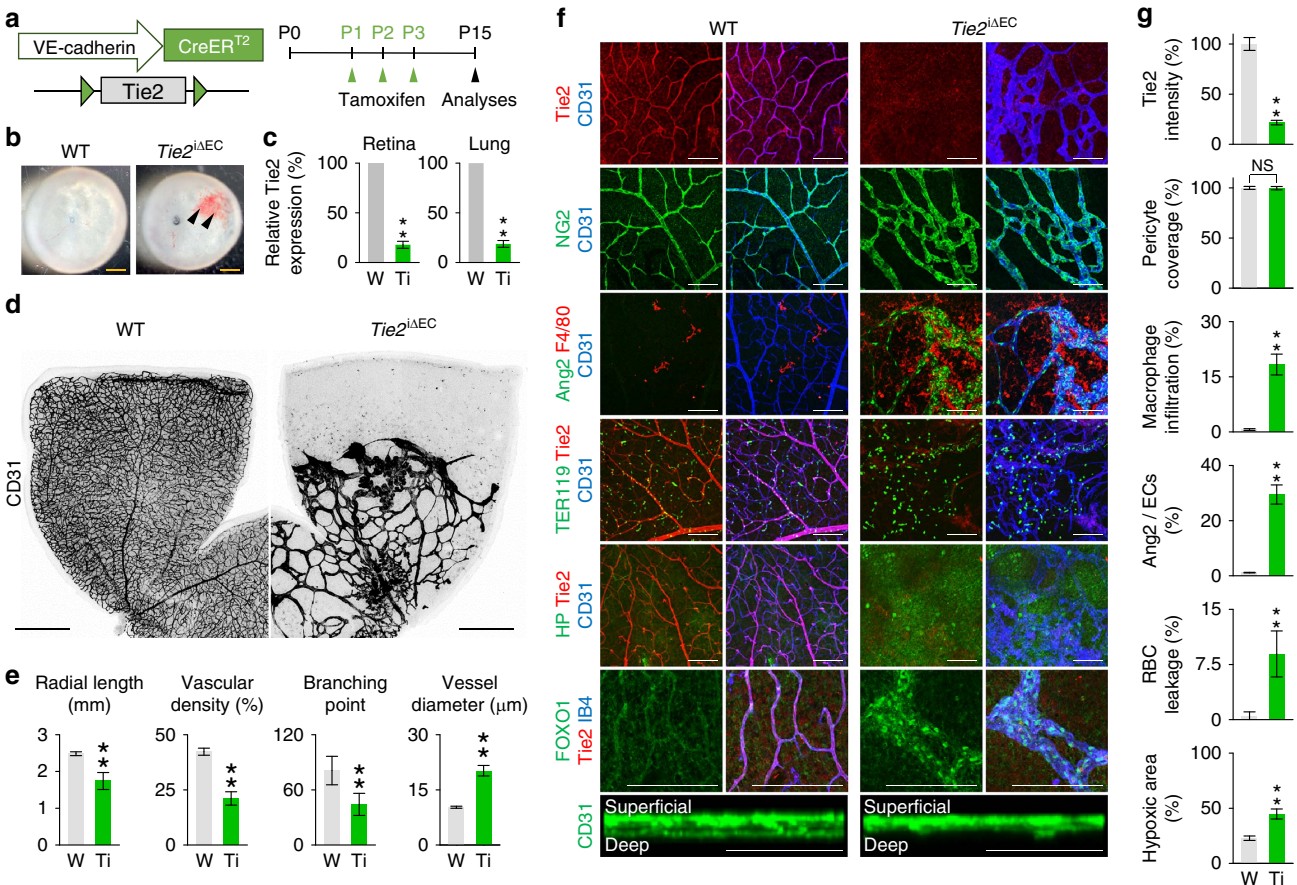

**Figure 5 | Tie2 is indispensable for retinal vascular growth and BRB maturation.** (**a**) Diagram depicting the experiment schedule for EC-specific depletion of Tie2 in retinal vessels from P1 and their analyses at P15 using $Tie2^{i\Delta EC}$ mice. (**b**) Images of inner surface of retinal cup of WT and $Tie2^{i\Delta EC}$ mice. Hemorrhages (arrowheads) are seen in $Tie2^{i\Delta EC}$ mice. (**c**) Relative levels of Tie2 mRNA in the lysates of retina and lung of $Tie2^{i\Delta EC}$ mice (Ti) mice versus WT mice (W). (**d,e**) Images of $CD31^+$ vessels and comparisons of indicated parameters in W ($n=6$) and Ti ($n=6$). (**f**) Images of retinal vascular phenotypes in W and Ti. Note that enlarged $IB4^+$ or $CD31^+$ vessels with adequate $NG2^+$ pericyte coverage have scanty expression of Tie2, while high expression of Ang2 by vessels with profound F4/80 macrophage infiltration, $TER119^+$ RBC leakage, severe hypoxia with impaired deep vessel formation, and evident nuclear localization of FOXO1 are detected in retinas of $Tie2^{i\Delta EC}$ mice. (**g**) Comparisons of indicated parameters in W ($n=6$) and Ti ($n=6$). Error bars represent mean $\pm$ s.d. $^{**}P<0.01$ versus WT by Mann–Whitney $U$ test. Scale bars, 100 μm (white) and 500 μm (yellow, black).

network formation with engorgement, impaired radial growth, severe leakage, and retinal detachment (Fig. 6b–g). However, these vascular impairments in radial growth, vascular density and diameter, RBC leakage, retinal detachment, macrophage infiltration, and Ang2 upregulation by APB5 were greatly ameliorated in $Foxo1^{i\Delta EC}$ mice (Fig. 6b–g).

To test whether the role of Ang2 is substantial in the breakdown of BRB maturation, we additionally depleted Ang2 in the ECs at P5 after crossing $Ang2^{flox/flox}$ mouse[41] with $Pdgfb^{i\Delta EC}$ mice ($Ang2:Pdgfb^{i\Delta EC}$ mice) and examined retinas at P12 (Fig. 7a). Micro-aneurysm formation, macrophage infiltration, and vascular leakage were greatly reduced in $Ang2:Pdgfb^{i\Delta EC}$ mice compared with $Pdgfb^{i\Delta EC}$ mice (Fig. 7b,c). Moreover, administration of Ang2-blocking antibody[42] also markedly attenuated micro-aneurysm formation, vascular enlargement, and RBC leakage upon APB5 administration (Fig. 7d–f). Although it was significant, genetic deletion of Ang2 or Ang2 blockade showed milder effects on relieving the BRB impairments compared with genetic deletion of FOXO1 (Figs 6 and 7). Thus, enhanced FOXO1 transcriptional activity upon inadequate pericyte attachment seems to be substantially responsible for deteriorating BRB maturation by upregulating several vascular destabilization factors including Ang2 (refs 29–31).

**Minimal effect of pericyte depletion on BRB in adult retina.** To uncover the role of pericytes in stabilized retinal vessels, we first depleted PDGF-B in the ECs of $Pdgfb^{i\Delta EC}$ mice by administration of tamoxifen during adulthood (Supplementary Fig. 12a). However, the depletion did not cause pericyte detachment or loss from the stabilized retinal vessels and led to no significant changes in the retinal vessels compared with those in WT mice (Supplementary Fig. 12b–f). Moreover, intra-vitreal administration of ABP5 (5 μg) into the eye of adult mice did not induce any apparent changes in the retinal vessels (Supplementary Fig. 13), implying that PDGF-B/PDGFRβ signalling is not required to maintain EC–pericyte interaction for BRB integrity during adulthood.

Second, we specifically ablated pericytes in the adult retinal vessels using $DTA^{i\Delta PC}$ mice, which were generated by crossing $PDGFR\beta$-$Cre$-$ER^{T2}$ mice with inducible ROSA26-diphtheria toxin A (DTA) mice[43] (Fig. 8a). Induction of DTA by tamoxifen had greatly ablated $PDGFR\beta^+$ and $NG2^+$ pericytes and $SMA^+$ SMCs in the retinal vessels of $DTA^{i\Delta PC}$ mice 2 weeks later (Fig. 8b–e). Surprisingly, no apparent changes in vascular density or diameter in the three vascular plexus layers and no apparent vascular leakage were detected in the retinas of $DTA^{i\Delta PC}$ mice compared with those of WT mice (Fig. 8b–e). Similarly, no apparent leakage was detected in the brain, but

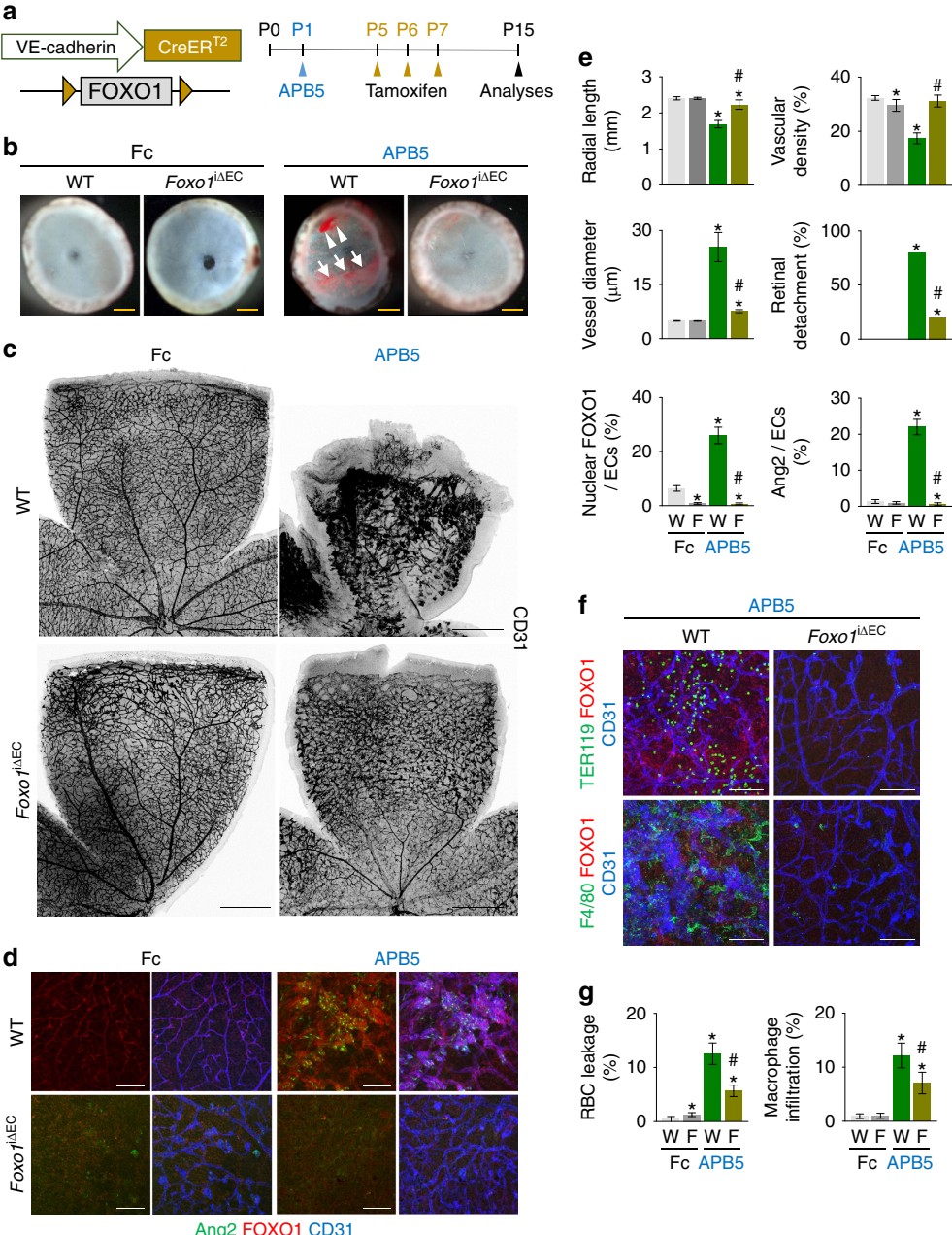

**Figure 6 | FOXO1 inactivation attenuates BRB impairments caused by lack of pericytes.** (**a**) Diagram depicting the experiment schedule for APB5 treatment at P1 and EC-specific depletion of FOXO1 (*Foxo1*[iΔEC]) from P5, and their analyses at P15. (**b**) Images of inner surface of retinal cup of WT and *Foxo1*[iΔEC] mice treated with either Fc of IgG (Fc) or APB5. Note that profound retinal hemorrhage (arrowheads) and retinal detachment (arrows) are detected in WT mice treated with APB5, but these phenotypes are largely attenuated in *Foxo1*[iΔEC] mice treated with APB5. (**c,d**) Images of CD31[+] retinal vessels, and expressions and localizations of Ang2 and FOXO1 in CD31[+] retinal vessels. Distinct retinal features such as disorganized and irregular enlarged vessels and less developed radial growth in WT mice treated with APB5 are largely ameliorated in *Foxo1*[iΔEC] mice treated with APB5. (**e**) Comparisons of indicated parameters in each group (*n* = 6). (**f,g**) Images and comparisons of TER119[+] RBC leakage and F4/80[+] macrophage infiltration in retinas of each group (*n* = 6). Error bars represent mean ± s.d. *$P < 0.05$ versus Fc-treated WT, #$P < 0.05$ versus APB5-treated WT by Kruskal–Wallis test. Scale bars, 100 μm (white) and 500 μm (yellow, black).

profound leakage was found in the lung and skin of DTA[iΔPC] mice (Supplementary Fig. 14a–c), implying that the role of pericytes in adult blood vessels might be organ specific. Even though we elevated systemic blood pressure up to 140 mm Hg (systolic pressure) for 3 weeks by sustained administration of angiotensin II using an osmotic mini pump, vascular leakage was not detected in the retinal vessels of DTA[iΔPC] mice (Supplementary Fig. 14d–f). By intra-vitreal administration of

tamoxifen into one side of the eye, we overcame the limitation of observation time period in DTA[iΔPC] mice due to early death (~3–4 weeks). Despite ~50% pericyte ablation, no apparent vascular remodelling and leakage were evident at 8 weeks later (Supplementary Fig. 15), which was the longest time period for observing the retinas of DTA[iΔPC] mice. On the other hand, impaired retinal vascular maturation together with BRB disruption, which were similar to the findings of *Pdgfb*[iΔEC]

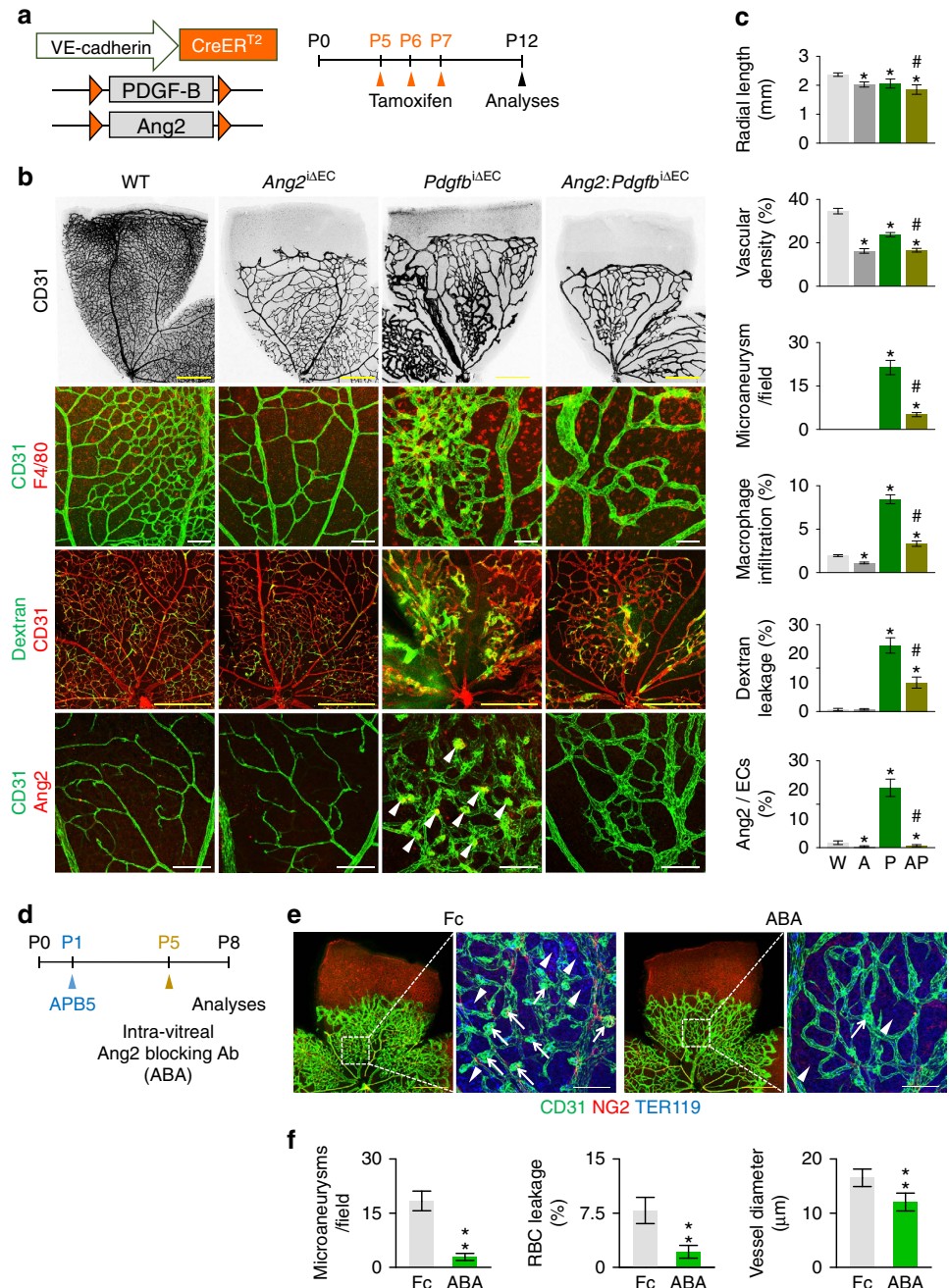

**Figure 7 | Ang2 blockade ameliorates pericyte loss-induced BRB impairments.** (**a**) Diagram depicting the experiment schedule for EC-specific depletion of PDGF-B and Ang2 in retinal vessels from P5 and their analyses at P12 in WT (W), *Ang2*[iΔEC] (A), *Pdgfb*[iΔEC] (P), and *Ang2:Pdgfb*[iΔEC] (AP) mice. (**b**) Images of CD31[+] retinal vessels, F4/80[+] macrophage infiltration, FITC-conjugated dextran (2,000 kDa) leakage, and expression and localization of Ang2 in each group. Note that abnormalities such as increased macrophage infiltration, vascular leakage, or microaneurysm (arrowheads) formation in *Pdgfb*[iΔEC] mice are greatly reduced in *Ang2:Pdgfb*[iΔEC] mice. (**c**) Comparisons of indicated parameters in each group (n = 6). Error bars represent mean ± s.d. *P < 0.05 versus W, #P < 0.05 versus P by Kruskal–Wallis test. (**d**) Diagram depicting the experiment schedule for intravitreal injection of Ang2 blocking antibody (ABA) at P5 in WT mice that were treated with APB5 at P1, and their analyses at P8. (**e,f**) Images and comparisons of CD31[+] vessels, NG2[+] pericytes, microaneurysms (arrows), and TER119[+] RBC leakages (arrowheads) in mice treated with Fc (n = 6) or ABA (n = 6). Error bars represent mean ± s.d. **P < 0.01 versus Fc by Mann–Whitney U test. Scale bars, 100 μm (white) and 500 μm (yellow).

mice, were observed in DTA[iΔPC] mice during postnatal development (Supplementary Fig. 16). These findings indicate that the regulation of pericytes on BRB could be dependent on stability of the vessels, and a sudden pericyte loss from the stabilized retinal vessels is insufficient to induce a significant alteration in the integrity of retinal vessels and

BRB, and additional insults may be required to mimic the hallmarks of DR.

**VEGF-A is required to recapitulate the phenotypes of DR.** Given that highly increased VEGF-A is a key factor for

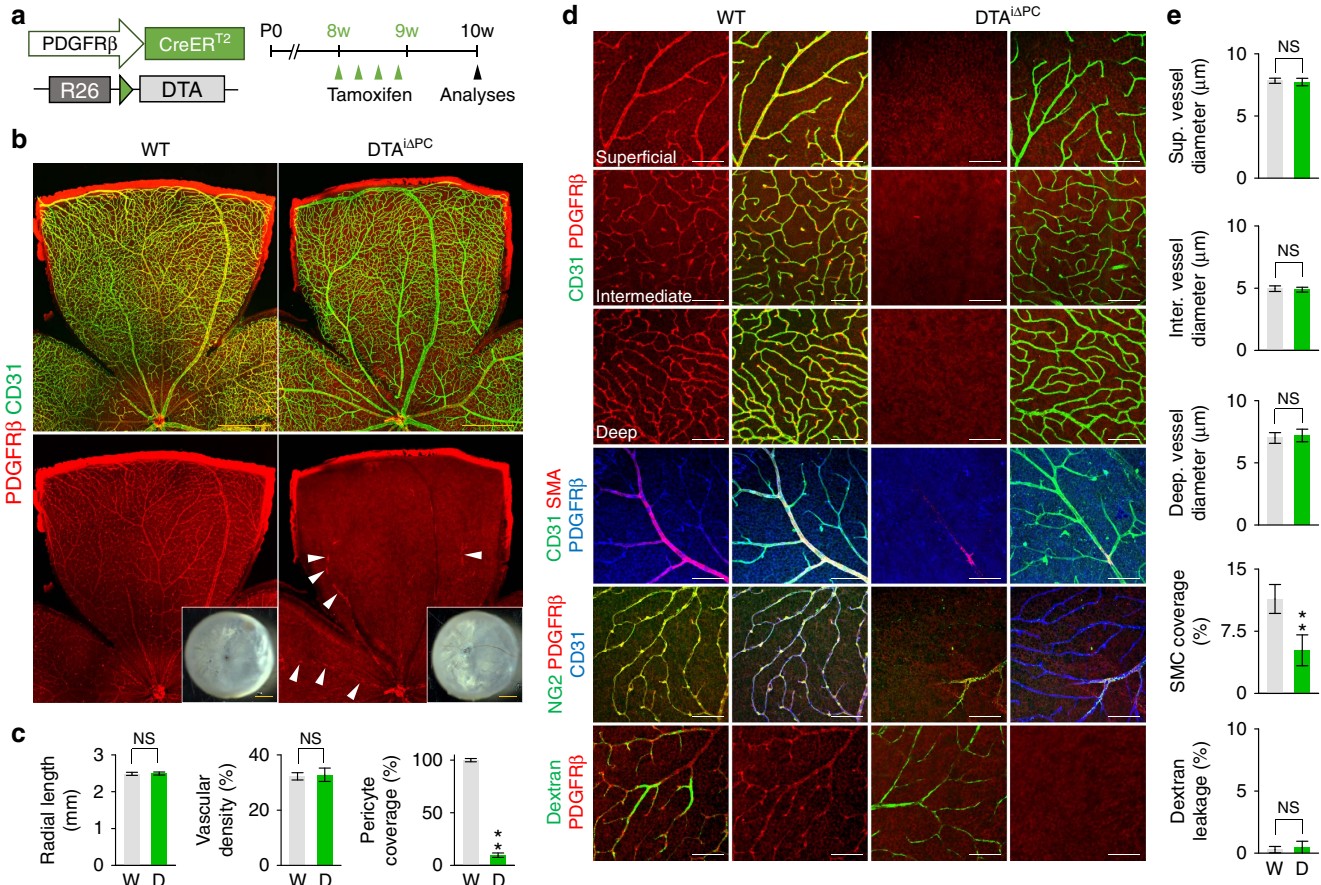

**Figure 8 | Pericyte-free retinal vessels of adults are resistant to vascular leakage.** (**a**) Diagram depicting the experiment schedule for selective loss of pericytes in retinal vessels of 8-week-old mice and their analyses at 2 weeks later using DTA$^{i\Delta PC}$ mice. (**b**) Images of CD31$^+$ vessels and PDGFRβ$^+$ pericyte coverage in WT and DTA$^{i\Delta PC}$ mice. Only a small number of pericytes (arrowheads) remained in retinal arteries of DTA$^{i\Delta PC}$ mice. Right lower panels, no visible hemorrhage is found in inner surface of retinal cups in both mice. (**c**) Comparisons of indicated parameters in WT (W; $n=6$) and DTA$^{i\Delta PC}$ (D; $n=6$) mice. (**d,e**) Images and comparisons of CD31$^+$ vessels and PDGFRβ$^+$ pericyte coverage in three layers, SMA$^+$ SMCs surrounding arteries, NG2$^+$ pericyte coverage, and dextran (70 kDa) leakage in retinas in WT (W, $n=6$) and DTA$^{i\Delta PC}$ (D, $n=6$) mice. Error bars represent mean ± s.d. **$P<0.01$ versus WT by Mann–Whitney $U$ test. NS, non-significant. Scale bars, 100 μm (white) and 500 μm (yellow).

destabilization, abnormal growth and permeability, and inflammation in the retinal vessels of DR[17,44,45], we administered VEGF-A (1 μg) intra-vitreally into the eye on one side of the pericyte-detached adult DTA$^{i\Delta PC}$ mice, while bovine serum albumin (BSA) (1 μg) was intra-vitreally administered into the contralateral eye as a control in the same mouse (Fig. 9a). Intriguingly, the VEGF-A–treated retinas of DTA$^{i\Delta PC}$ mice exhibited retinal hemorrhage, tortuous and enlarged vessels, and markedly increased Ang2, VEGFR2, FOXO1 nuclear localization and vascular leakage but decreased Tie2, while the BSA-treated retinas of those mice showed no apparent abnormalities in vascular structure or permeability (Fig. 9b,c; Supplementary Fig. 17). In comparison, minor to moderate increases in Ang2 and FOXO1, but not in Tie2, were detected in retinas of DTA$^{i\Delta PC}$ mice treated with BSA, whereas increases in VEGFR2 and vascular leakage were observed in retinas of WT mice treated with VEGF-A (Fig. 9b,c; Supplementary Fig. 17). Thus, pericyte is a gatekeeper regulating FOXO1 that protects retinal vessels against stresses, injuries, and insults including excess VEGF-A.

To test whether Tie2 activation could reduce vascular leakage induced by VEGF-A in pericyte-free retinal vessels, we treated the DTA$^{i\Delta PC}$ mice with daily intra-peritoneal administration (10 mg kg$^{-1}$) of Fc or ABTAA[42] for 4 days just after the VEGF-A injection and analysed their retinal vessels at post-injection

day 4 (Fig. 10a). Compared with Fc, ABTAA reduced vascular leakage and macrophage infiltration without restoration of pericyte coverage onto the vessels (Fig. 10b,c), which is similar to the findings reported previously[11]. Thus, Tie2 activation could ameliorate BRB disintegration against VEGF-A stimuli in the absence of pericytes, preventing retinal damages during DR progression.

## Discussion

In this study, we demonstrate that the PDGF-B/PDGFRβ signalling is not required to maintain EC–pericyte interaction for BRB integrity during adulthood, although it is indispensable in formation and maturation of BRB through active recruitment of pericytes onto the growing retinal vessels. Moreover, strikingly, a sudden pericyte loss did not significantly disturb the stabilized, quiescent adult retinal vessels and BRB for some time. Furthermore, the pericyte-free retinal vessels were resistant to vascular leakage/hemorrhage and disruption unless the ECs were damaged by noxious stimulus, such as excess of VEGF-A. Pericyte dropout is one of the major hallmarks of DR[15–17]. A long-standing question has been whether pericyte dropout is a primary causative factor, a secondary consequence of micro-vessel abnormality, or both in the pathogenesis of DR. Our findings clearly demonstrate that pericyte dropout is not

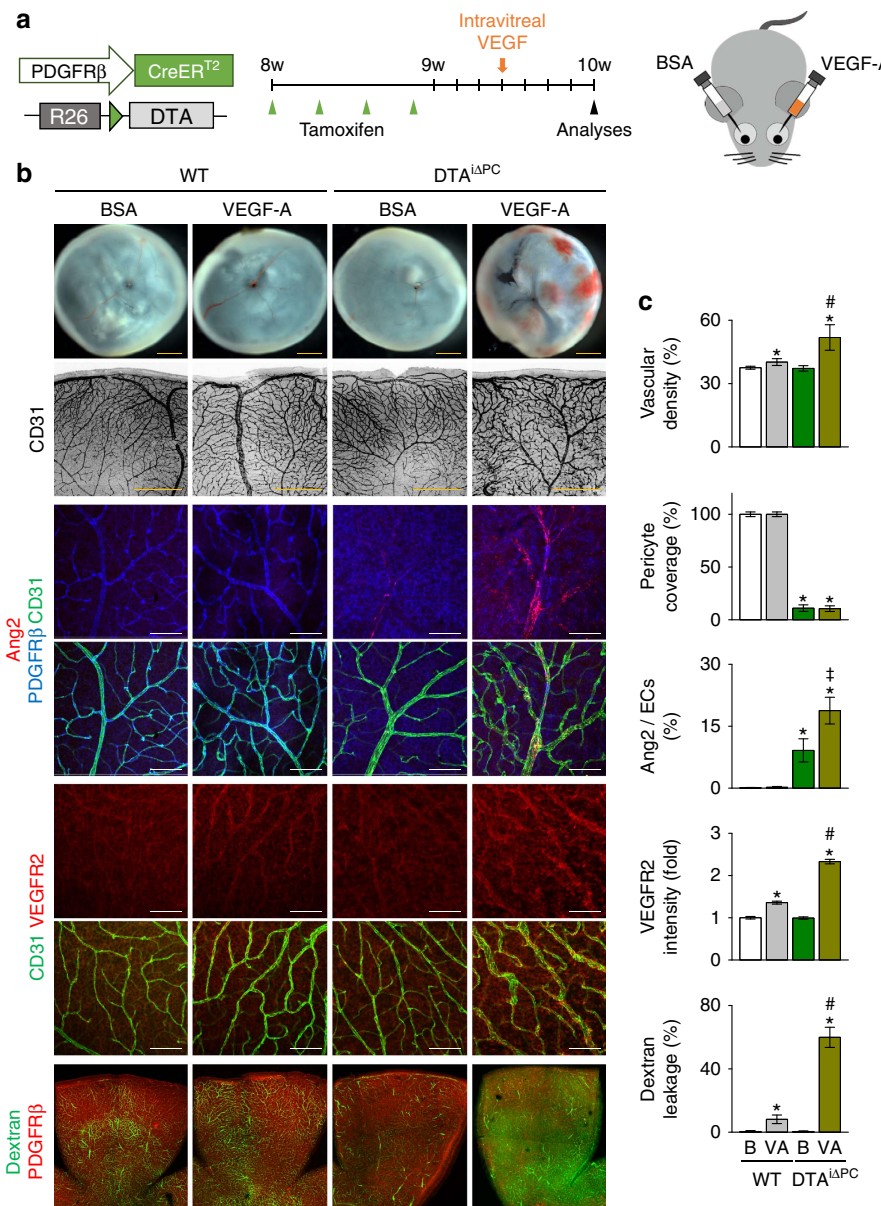

**Figure 9 | Additional VEGF-A is required to recapitulate the phenotypes of DR.** (**a**) Diagram depicting the experiment schedule for selective loss of pericytes in retinal vessels using adult DTA$^{i\Delta PC}$ mice, intra-vitreal administration of VEGF-A (1 µg) into one eye and BSA (1 µg) into the contralateral eye, and analyses at 4 days after the administration. (**b,c**) Images and comparisons of inner surface of retinal cup, CD31$^+$ vessels, PDGFRβ$^+$ pericyte coverage, expressions of Ang2 and VEGFR2 in CD31$^+$ vessels, and dextran (70 kDa) leakage in retinas of WT and DTA$^{i\Delta PC}$ mice that were intra-vitreally treated with BSA (B) or VEGF-A (VA). Each group, n = 6. Error bars represent mean ± s.d. *$P < 0.01$ versus WT treated with B, $^{\ddagger}P < 0.01$ versus DTA$^{i\Delta PC}$ treated with B, $^{\#}P < 0.01$ versus WT treated with VA by Kruskal–Wallis test. Scale bars, 100 µm (white) and 500 µm (yellow).

a direct causative factor for DR pathogenesis but rather is an essential factor that accelerate DR progression, further destabilizing retinal vascular ECs. In this regard, either preventing pericyte detachment or enhancing EC–pericyte interaction could be a potential therapeutic approach for ameliorating DR progression by maintaining BRB integrity and function.

In contrast to the prevailing concept[6,19–21], we provide compelling evidence that pericytes that surround retinal vessels do not express Ang1. Despite the lack of Ang1 expression by pericytes, globally Ang1-depleted neonatal mice show reduced vascular growth and pericyte coverage in the retina[37]. In the context of Tie2, we showed that endothelial Tie2 plays an

indispensable role in vascular development and BRB maturation. It is well-known that Ang1-induced Tie2 clustering is critical for its activation[19,46,47]. However, the activation of Tie2 can also be strongly induced by shear stress[48], which is an independent mode of action to Ang1-induced activation. These collectively imply that Ang1 may have derived from adjacent neurons or hematopoietic cells including platelets, which subsequently activate Tie2 or integrins for its roles in formation and maturation of the inner part of the BRB, or the abnormal vascular phenotypes after endothelial Tie2 deletion can be due to the absence of Tie2 activation induced by a blood flow-mediated shear stress. Mechanistically, the gate-keeping role of intact pericyte-Tie2/Akt signalling, which suppresses FOXO1-mediated

transcriptional activation of vascular destabilizing factors including Ang2 in ECs, seems central for the maintenance and protection of BRB integrity against injuries (Fig. 10d). The

mechanism by which pericyte coverage regulates Tie2 can be explained in two ways. First, Tie2 shedding (cleavage of Tie2 extracellular domain) in ECs could have occurred immediately

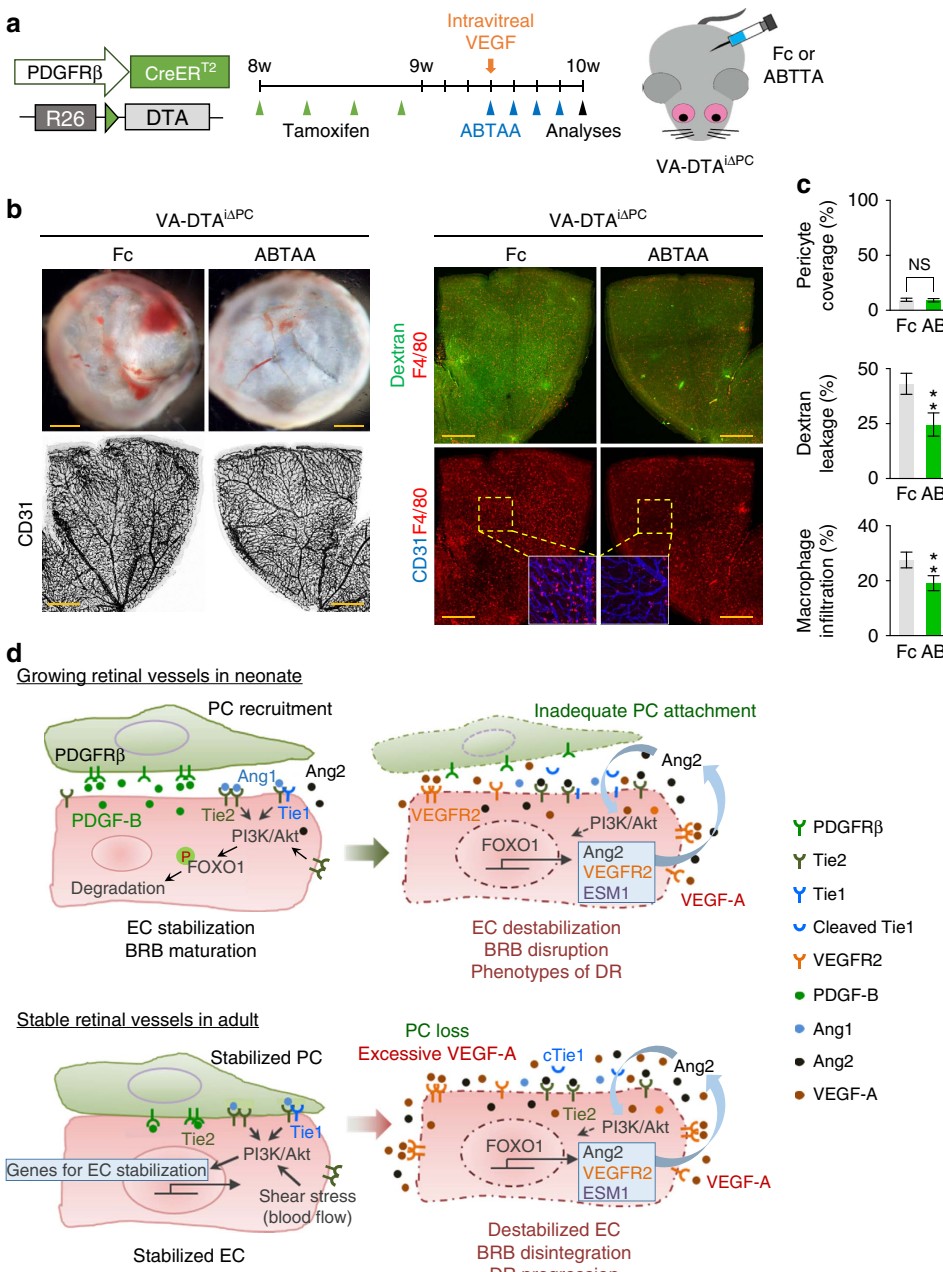

**Figure 10 | Tie2 activation lessens BRB disintegration in pericyte-free adult retina.** (**a**) Diagram depicting the experiment schedule for selective depletion of pericytes around retinal vessels by tamoxifen, intra-vitreal administration of VEGF-A (1 μg), and daily intra-peritoneal administration of ABTTA (20 mg kg$^{-1}$) for 4 days in adult DTA$^{i\Delta PC}$ mice. Analyses were performed 4 days after VEGF-A administration. (**b,c**) Images and comparisons of inner surface of retinal cup, CD31$^+$ vessels, dextran (70 kDa) leakage, and F4/80$^+$ macrophage infiltration in VA-DTA$^{i\Delta PC}$ mice treated with Fc (Fc; $n = 6$) or ABTAA (AB; $n = 6$). Boxed regions (dotted-lines) are magnified and presented in panels in the corner. Error bars represent mean ± s.d. ** $P < 0.01$ versus Fc by Mann–Whitney $U$ test. All scale bars, 500 μm (yellow). (**d**) Schematic diagrams depicting how inadequate pericyte coverage disintegrates BRB. In growing retinal vessels of neonates, proper pericyte recruitment through controlled PDGF-B/PDGFRβ signalling stabilizes ECs by activating the Tie2/Tie1-PI3 kinase-Akt signalling cascade, which leads to phosphorylation (P) and degradation of FOXO1 in ECs and promotes EC-pericyte interaction during BRB formation. Inadequate pericyte recruitment due to PDGFRβ signalling inhibition attenuates Tie2/Tie1-PI3 kinase-Akt signalling, which activates FOXO1-mediated transcription of Ang2, VEGFR2 and ESM1 in ECs. In stable retinal vessels of adult, several key factors including Tie2/Tie1- and shear stress-mediated activation of PI3 kinase-Akt signalling play key roles in maintenance of BRB integrity. Pericyte loss increases Ang2 and sensitivity to external noxious stimuli such as VEGF-A, leading to attenuated PI3 kinase-Akt signalling and enhanced FOXO1-mediated transcription of vascular destabilizing factors. In both cases with inadequate pericyte coverage around retinal vessels, increased Ang2 and Tie1 cleavage and reduced Tie2 further attenuated PI3 kinase-Akt signalling, leading to enhanced FOXO1-mediated transcription of vascular destabilizing factors in a positive feedback manner. These exacerbating processes eventually trigger further disintegration of BRB, manifesting DR phenotypes and accelerating DR progression.

after pericyte loss, because ECs are directly exposed to a bunch of activated proteases and growth factors such as VEGF-A. In fact, VEGF-A is known to induce Tie2 shedding via a phosphatidylinositol 3 (PI3) kinase/Akt-dependent pathway to modulate Tie2 signalling[49]. In addition, recent elegant studies[33,34,50] have revealed that inflammation-induced cleavage of the Tie1 extracellular domain leads to reduced availability and signalling of Tie2, which in turn further degrades vascular stability through FOXO1-mediated upregulation of Ang2 in the ECs of tracheal vessels. Thus, in agreement with these studies, inflammation induced by the BRB disruption after pericyte loss could also reduce functional Tie2 level and stimulate FOXO1 activation. Nevertheless, unlike FOXO1 and Ang2, the change in Tie2 upon pericyte depletion was different between growing retinal vessels and stable adult retinal vessels, implying that regulation of Tie2 by pericytes could also be dependent on vessel stability as well as its microenvironment, such as inflammation and hypoxia. PI3 kinase/Akt intracellular signalling is the main downstream pathway for Tie2 activation in ECs[19,47]. While PI3 kinases play key roles in the maintenance of vascular integrity and barrier function[51,52], Akt also plays a vital role in maintaining vascular integrity against VEGF-induced impairments and increased permeability in blood vessels[53–55]. In fact, Akt acts as a negative regulator of FOXO1 by promoting its phosphorylation and degradation[19,29]. In addition, a previous study has shown that FOXO1 activity is induced together with Akt inactivation in primary cultured ECs of microvasculature under diabetic conditions[56]. Thus, pericyte loss-induced FOXO1 activation could be due to the reduced Tie2-mediated PI3 kinase/Akt signalling.

Upregulation of Ang2 expression and enhanced secretion of Ang2 in ECs are highly reactive processes that occur in any unstable and stimulatory conditions, including acute inflammation, chronic hyperglycemia, cerebral cavernous malformation and excess of vascular stimulating factors and cytokines, which definitely contribute to further exacerbation of vascular destabilization in a positive feedback manner[57–60]. Indeed, our findings clearly show that pericyte loss induces FOXO1-mediated transcriptional activation of the genes related to vascular destabilization such as Ang2, ESM1 and VEGFR2 (refs 25,27,29,32), which sensitizes ECs to VEGF-A and further disintegrates BRB by a positive feedback system in retinal vessels (Fig. 10d)[25,29,61,62]. In fact, Ang2 per se is a positive factor for pericyte dropout in animal models of DR[23,63], although its role in pericyte dropout could be overridden in a context-dependent manner as was the case in the Tie2[iΔEC] mice in this study (Fig. 5f). Ang2 is also detected in high levels in the vitreous fluid of patients with DR[64]. Moreover, a recent study[60] revealed that cerebral cavernous malformation 3 deficiency in brain ECs enhances the exocytosis and secretion of Ang2, which leads to destabilization of BBB and BRB together with pericyte dropout. In the present study, our approaches with genetic deletion of Ang2 or FOXO1, or administration of Ang2 blocking antibody highlight the critical roles of both Ang2 and FOXO1 as negative regulators of EC stability in the absence of pericytes. However, neither genetic deletion nor antibody administration fully mitigated the vascular leakage caused by inadequate pericyte coverage. There could be several reasons for this partial rescue. First, there could be unidentified molecular actions or pathways that are responsible for the pericyte depletion-induced vascular leakage but independent of the action of FOXO1 or Ang2. Second, the deletion efficiency may not have been 100% to fully inactivate the actions of FOXO1 or Ang2; similarly, Ang2 blocking antibody would not have perfectly acted on all existing Ang2 due to issues with drug delivery and binding efficiencies. Nevertheless, as we and others have demonstrated[65–67], blockade

of Ang2 and VEGF-A along with Tie2 activation could be beneficial in ameliorating BRB breakdown and DR progression.

Together, this study delineates the plastic roles of PDGF-B/PDGFRβ signalling in EC–pericyte interaction depending on vessel status and identifies the critical role of the Tie2–FOXO1–Ang2 loop in pericyte dropout during BRB breakdown in the context of DR, providing clues regarding its pathogenesis and treatment.

## Methods

**Experimental animals.** Animal care and experimental procedures were performed under the approval from the Institutional Animal Care and Use Committee (No. KA2015-15) of KAIST. Specific pathogen-free (SPF) C57BL/J, Pdgfb[flox/flox] (C57BL/J background)[9], and R26-tdTomato (C57BL/J background) were purchased from the Jackson Laboratory. VE-Cadherin-Cre-ER[T2] (ref. 24), Ang1[flox/flox] (ref. 37), Ang2[flox/flox] (ref. 41), Tie2[flox/flox] (ref. 27), Foxo1[flox/flox] (ref. 40), Ang1-GFP knock-in[36], PDGFRβ-Cre-ER[T2] (ref. 38), and ROSA26-DTA[43] mice were transferred, bred in our SPF animal facilities at KAIST, and used after more than 10 generations of back-crossing into C57BL/J background. Male and female mice were not distinguished in neonatal mice (P5 to P15), and male mice were used in adult (after P28) experiments. To induce Cre activity in the Cre-ER[T2] mice, tamoxifen was given at the indicated days with the following dosages and schedules: for neonatal mice, 50 μg of 4-hydroxy (4-OH) tamoxifen (Sigma, H7904) was injected into the stomach or subcutaneously daily from P1 to P3, or 100 μg daily from P5 to P7; for OIR mice, 200 μg of tamoxifen (Sigma, T5648) was injected i.p. daily from P12 to P14; for adult mice aged over 1 month, 2 mg of tamoxifen was injected i.p. for 4 consecutive days from the indicated time point. To induce Cre activity in one eye, 1 μl of 4-OH tamoxifen dissolved in DMSO (10 μg μl[−1]) was injected into the vitreous cavity in one eye using Nanoliter 2000 micro-injector (World Precision Instruments) fitted with a glass capillary pipette. As control, 1 μl of DMSO was injected in the same manner to the contralateral eye. Cre-ER[T2] positive but flox/flox-negative mice among the littermates were defined as wild type (WT) mice for each experiment. To increase blood pressure, Angiotensin II (AT II, Calbiochem) was infused at a rate of 1.44 mg kg[−1] min[−1] for 3 weeks through an osmotic pump (Alzet) implanted in the subcutaneous area. All mice were bred and housed in our SPF facilities and fed with free access to a standard diet (PMI Lab diet) and water. Mice were handled in accordance with the ARVO (Association for Research in Vision and Ophthalmology) Statement for the Use of Animal in Ophthalmic and Vision Research.

**Generation of OIR mouse model.** The OIR mouse model was generated as previously reported[37]. Briefly, P7 mice were exposed to 75% oxygen in a hyperoxic chamber (ProOx Model 110, BioSpherix, Lacona, New York) for 5 days with their nursing mothers and then returned to room air for 5 days. Retinas were harvested at P17.

**Treatment of antibody or recombinant protein.** For in vivo inhibition of PDGFRβ signalling, 50 μg of APB5 (5 mg ml[−1]) was injected subcutaneously to neonatal mice at P1, or 5 μg of APB5 was injected intravitreally for adult mice. To administer the antibody or recombinant protein into the vitreous cavity, indicated amount of reagent in 1 μl was injected into the vitreous cavity.

**Histological analyses.** Immunofluorescence (IF) staining of whole-mounted retinas and retinal sections were performed as previously described[37]. Briefly, eyeballs were enucleated and fixed in 4% paraformaldehyde (PFA) for 20 min at room temperature (RT). For IF staining of retinal section, samples were dehydrated in 20% sucrose solution overnight, embedded in tissue freezing medium (Leica), and cut into 14 μm section. For IF staining of whole-mounted retina, after preparing the retina from eyeball, the retinas were additionally fixed in 1% PFA for 1 h at RT. Samples were blocked with 5% donkey (or goat) serum in PBST (0.3% Triton X-100 in PBS) and then incubated in blocking solution with isolectin B4 (Sigma, L2140) or following antibodies at 4 °C overnight: hamster anti-CD31 monoclonal (clone 2H8, Millipore, MAB1398Z, 1:500); rat anti-PDGFRβ monoclonal (clone APB5, eBioscience, NC0091961, 1:200); rabbit anti-NG2 monoclonal (132.39, Millipore, MAB5384, 1:200); human anti-Ang2 monoclonal (clone 4H10, 1:200)[42]; rat anti-TER119 monoclonal (clone TER-119, BD Pharmingen, 561033, 1:200); rat anti-F4/80 monoclonal (BM8, eBioscience, 14-4801-81, 1:200); rabbit anti-FOXO1 monoclonal (clone C29H4, Cell signaling, 2880, 1:200); rat anti-VE-cadherin monoclonal (clone 11D4.1, BD Pharmingen, 555289, 1:200); FITC-conjugated mouse anti-SMA monoclonal (clone 1A4, Sigma, F3777, 1:1,000); rabbit anti-phosphohistone H3 (PH3) polyclonal (Millipore, 05-806, 1:200); rabbit anti-caspase-3 polyclonal (R&D, AF835, 1:200); goat anti-Tie2 polyclonal (R&D, AF762, 1:200); goat anti-Tie1 polyclonal (R&D, AF619, 1:200); rabbit anti-ZO1 polyclonal (Thermo Fisher Scientific, 61-7300, 1:200); chicken anti-GFP polyclonal (Abcam, ab13970, 1:200); goat anti-ESM1 polyclonal (R&D, AF1999, 1:100); goat anti-VEGFR2 polyclonal (R&D, AF644, 1:200) antibody. After several washes, the samples were incubated for 2 h at RT with

FITC-, Cy3-, or Cy5-conjugated streptavidin (BD pharmingen, 1:1,000) or the following secondary antibodies: FITC-, Cy3-, or Cy5-conjugated anti-hamster IgG, anti-rat IgG, anti-rabbit IgG, anti-human IgG, anti-chicken IgG, and anti-goat IgG (Jackson ImmunoResearch, 1:1,000). To detect hypoxic area in retinas, Hypoxyprobe-1 ($60\,mg\,kg^{-1}$; solid pimonidazole HCL, Natural Pharmacia International, Belmont, Massachusetts) was injected i.p. 1 h before enucleation. The retinas were then harvested and stained with FITC-conjugated anti-Hypoxyprobe antibody. Images of all samples were obtained using a Zeiss LSM 780 confocal microscope (Carl Zeiss). *In situ* hybridization (ISH) for PDGF-B on the whole-mounted retina was performed as previously described[68]. Briefly, after short fixation of eyeball with 2% PFA, retinas were harvested, digested with proteinase K, and postfixed for 5 min in 4% PFA/1% glutaraldehyde solution. Then, the retinas were hybridized with digoxigenin (DIG)-labelled PDGF-B cRNA probes (811-1505 of NM_011057) at $65\,°C$ overnight and were subsequently incubated with anti-DIG-AP antibody (Roche, 11 093 274 910) at RT overnight. Colour was developed by reacting with nitroblue tetrazolium chloride (Roche, 1 087 479)/5-bromo-4-chloro-3-indoyl phosphate (Roche, 1585 002). After extensive washes with PBS, for visualization of the retinal vessels, retinas were IF stained with a rabbit anti-type IV collagen polyclonal antibody (Cosmo bio, LB-1403, 1:200) and a Cy3 conjugated anti-rabbit secondary antibody (Jackson ImmunoResearch, 1:1,000). The images of ISH and IF were taken separately and merged after pseudo-colouring using Adobe Photoshop CS software (Adobe).

**Analyses of vascular leakage and perfusion.** Vascular leakage was analysed by i.v. injecting $50\,\mu l$ of FITC-conjugated dextran of indicated size ($25\,mg\,ml^{-1}$, 70 kDa/2,000 kDa, Sigma-Aldrich) 30 min before organ harvest, or $100\,\mu l$ of Evans blue (EB) (2% in PBS; Sigma). To quantify EB leakage in different organs, the mice were perfused with PBS and organs were harvested and weighed 24 h after EB injection. Then the organs were homogenized and incubated in formamide for 24 h at $55\,°C$. Supernatants were collected and EB contents were measured at the absorbance of 620 nm. EB concentration per weight of wet organs were calculated from a standard curve of EB in formamide, and normalized by control and presented as fold difference. For analysis of vascular perfusion, $50\,\mu l$ of DyLight 488-labelled tomato lectin (Vector laboratory, DL-1174) was i.v. injected before enucleation. Intracardiac perfusion with PBS was performed before mice sacrifice to remove circulating lectin.

**Morphometric analyses.** Morphometric analyses of the retina were performed using ImageJ software (http://rsb.info.nih.gov/ij) or ZEN 2012 software (Carl Zeiss). Radial length of the retinal vessels was measured as the shortest distance from the optic disc to the peripheral vascular front in each quadrant of the retina and averaged. Retinal vascular density was measured as $CD31^+$ or $IB4^+$ retinal vessel area divided by total measured area of the retina and presented as a percentage. Branching point was measured manually in six $0.18\,mm^2$ fields located between an artery and a vein in each retina and averaged. Number and length of filopodias were examined in four $0.02\,mm^2$ areas of vascular front in each retina and averaged. Vessel diameters were averaged among 10 consecutive $CD31^+$ or $iB4^+$ retinal vessels between an artery and a vein located within $500\,\mu m$ from the optic disc. Numbers of $PH3^+$ proliferating EC or sprouting EC were measured in four $0.18\,mm^2$ fields of vascular front per retina and averaged. Numbers of caspase3$^+$ EC were measured in six $0.18\,mm^2$ fields of vascular plexus between an artery and a vein per retina and averaged. Pericyte coverage was calculated as $PDGFR\beta^+$ or $NG2^+$ area divided by $CD31^+$ or $IB4^+$ area, which was then normalized by the average of those of control mice. SMC coverage was calculated as $SMA^+$ area divided by $CD31^+$ area. Detached SMC was calculated as area of $SMA^+$ SMC not covering $CD31^+$ retinal vessels divided by total $SMC^+$ area. RBC leakage were measured as RBC-stained area outside the vessels divided by retinal area. Macrophage infiltration was measured as $F4/80^+$ area divided by retinal area. RBC leakage and macrophage infiltration were measured in six random $0.18\,mm^2$ fields per retina and averaged. To measure the hypoxic status of the retina, FITC-labelled Hypoxyprobe$^+$ area was measured and divided by total retinal area and presented as a percentage. Micro-aneurysm was defined as ballooning or outpouching of $CD31^+$ or $IB4^+$ retinal vessels without connection with deep vascular plexus and counted in four $0.18\,mm^2$ fields per retina and averaged. NVT area and avascular areas in OIR model were measured using the Lasso tool of Adobe Photoshop software as previously described[37] and divided by total retinal area. Staining intensities were measured in four representative areas of retinal capillary vessels or, if indicated, other vascular regions in each retina and averaged. For comparison, the values were normalized by the background signals in non-vascularized areas, and their ratio were normalized by control and presented as fold or percentage. We regarded the round and condensed signal of FOXO1 immunofluorescence staining in the retinal vessels as a nuclear-localized activated form of FOXO1, in accordance to a recently published report[33]. Then, area of the nuclear-localized FOXO1 per $CD31^+$ EC area was calculated using same threshold values using ImageJ software and presented as 'nuclear FOXO1/ECs (%)'. Similarly, to quantify Ang2 or ESM1 expression, their stained area per $CD31^+$ EC area was calculated and presented as 'Ang2/ECs (%)' or 'ESM1/ECs (%)'. If needed, their values were normalized by control and presented as fold change.

**Quantitative real-time RT-PCR.** Total RNA was extracted using RNeasy Plus Mini kit (Qiagen) according to the manufactuer's instruction. Total RNA was reverse transcribed into cDNA using GoScriptTM Reverse Transcription Kit (Promega). Then, quantitative real-time PCR was performed using FastStart SYBR Green Master mix (Roche) and Bio-rad S1000 Thermocycler with the indicated primers (Supplementary Tables 1 and 2). GAPDH was used as a reference gene and the results were presented as relative expressions to control.

**Electroretinogram.** Electroretinogram (ERG) was recorded to assess the retinal neuronal function at indicated time point. Mice were either dark- or light-adapted for 12 h before ERG monitoring, anesthetized with $15\,mg\,kg^{-1}$ ketamine and $7\,mg\,kg^{-1}$ xylazine, and placed on a heating pad to maintain body temperature. After pupil dilatation using tropicamide 0.5% (Mydrin P, Taejoon, Korea), cornea was placed on gold-plated objective lens and silver-embedded needle electrodes were placed at forehead (reference) and tail (ground). The ERG stimulus and recording were performed using 'LabscribeERG v3' software combined with 'Phoenix Micron IV' (Phoenix Research labs, California) retinal imaging microscope according to the manufacturer's instruction. To obtain scotopic (dark-adapted condition) a- and b-wave, a digital bandpass filter ranging from 0.3 to 1,000 Hz and stimulus ranging from $-2.2$ to 2.2 $\log(cd \cdot s\,m^{-2})$ were used. To yield photopic (light-adapted condition) b-wave, filter ranging from 2 to 200 Hz and stimulus ranging from 0.4 to 2.2 $\log(cd \cdot s\,m^{-2})$ with 1.3 $\log(cd \cdot s\,m^{-2})$ background were used. After averaging the signals, the amplitude of the a- and b-wave were presented by the 'LabscribeERG v3' software and used for analyses.

**Aortic ring assay.** Aortic ring assay was performed as previously described[35]. Briefly, aortas of the mice treated with tamoxifen as scheduled were harvested at the indicated time point. After overnight serum starvation in Opti-MEM (Life Technologies), aortas were cut into 1 mm thick segments, embedded in type I collagen gel (Millipore) in a 96-well plate, and incubated with Opti-MEM containing 2% FBS (Hyclone) and $30\,ng\,ml^{-1}$ VEGF-A (R&D) for 6 days. For immunostaining of aortic ring[35], aortic ring embedded in collagen gel was fixed in 4% PFA for 10 min on ice, blocked with 5% donkey (or goat) serum in PBST (0.3% Triton X-100 in PBS) for 1 h at RT, and incubated in blocking solution with primary antibodies at $4\,°C$ overnight. After several washes, the samples were incubated for 2 h at RT with secondary antibodies. Then, imaging was performed using Zeiss LSM 780 confocal microscope (Carl Zeiss). Number of sprouting vessel, pericyte coverage, and the signal densities of Ang2 and FOXO1 were quantified using ImageJ software.

**Cell culture and siRNA transfection.** HUVECs (Lonza, C2519A) were cultured in EGM2 medium (Lonza) supplemented with growth factor and serum. For silencing experiments, HUVECs were transfected with either Tie2 siRNA, FOXO1 siRNA, or control siRNA (Life Technologies) using liopfectamine RNAiMAX (Invitrogen) in Opti-MEM (Life Technologies) according to vender's instruction. After transfection, cells were incubated until it reached 80–90% confluency before harvest for transcript analysis.

**ELISA of VEGF-A and soluble Tie1 in vitreous fluid.** To collect vitreous fluid in the eye, first, eyes were enucleated and washed with sterile ice cold PBS. Then, PBS around the eyes was removed using an absorbing paper. After making a small slit with a blade through the retina, vitreous fluid was collected using Nanoliter 2000 micro-injector (World Precision Instruments) fitted with a glass capillary pipette. To measure the intravitreal levels of soluble Tie1 or VEGF-A, after 40-fold dilution of vitreous sample, mouse Tie1 ELISA kit (CUSABIO, CSB-E07391m) or mouse VEGF-A ELISA kit (R&D, MMV00) were used according to the manufacturer's instruction and measured using a Spectra MAX340 plate reader (Molecular Devices).

**Chromatin immunoprecipitation-qPCR.** $\sim 5 \times 10^6$ HUVECs were harvested and fixed with 11% formaldehyde solution and quenched by 2.5 M glycine. After washing with PBS 3 times, the cells were lysed and sonicated using a Bioruptor sonicator (Diagenode). Chromatin immunoprecipitation (ChIP) was performed using FOXO1 antibody (ab39670, Abcam, 1:100) and magnetic protein-G beads (Thermo Fisher Scientific). After several washes, reverser crosslink, DNA extraction, and qPCR were performed using input DNA and ChIP-DNA. H3K27ac HUVEC ChIP-seq data was obtained from GEO database (GSM733691)[69] and analysed by Integrative Genomic Viewer (Broad Institute)[70] and used to select expected enhancer regions of Ang2. The primer sequences for ChIP-qPCR are shown in Supplementary Table 3. eNOS promoter and eNOS coding sequence regions were used as positive and negative controls for FOXO1 binding[30].

**Statistical analyses.** Sample sizes were estimated by a power analysis ('TrialSize' package of R3.2.3 version) to detect a 20–40% difference ($\delta$) with a significance level ($\alpha$) of 0.05 and power ($\beta$) of 0.8. Animals or samples were not randomized during experiments and not excluded from analyses. The investigators were not blinded to group allocation during experiments and outcome analyses.

Values were presented as mean ± standard deviation (s.d.). Statistical significance was determined by the two-sided Mann–Whitney $U$ test between 2 groups or the Kruskal–Wallis test followed by Tukey's honest significant difference (HSD) test with ranks for multiple-group comparison. Statistical analysis was performed using PASW statistics 18 (SPSS). Statistical significance was set to $P < 0.05$.

**Data availability.** The authors declare that all relevant data of this study are included within the article and its Supplementary Information files. These are also available from the authors upon request.

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

## Acknowledgements

We thank Intae Park for proof reading of manuscript, Su Jin Seo and Mi-Jeong Kim for their technical assistances, Professor Yoshikazu Nakaoka (Osaka University) for providing *Ang1*<sup>flox/flox</sup> mice, Professor Ronald A. DePinho (University of Texas MD Anderson Cancer Center) for providing *Foxo1*<sup>flox/flox</sup> mice, Professor Richard M. Locksley (University of California San Francisco) for providing ROSA26-DTA mice, and Prof Akiyoshi Uemura (Nagoya City University) for providing APB5. This study was supported by the Institute of Basic Science funded by the Ministry of Science, ICT and Future Planning, Korea (IBS-R025-D1-2015, G.Y.K), and the Rita Allen Foundation and NIH R01HL132074 (L.D).

## Author contributions

D.Y.P., J.L., J.K., K.K., S.P.H., S.Y.H. and H.K. designed and performed the experiments and analysed the data; Y.K., H.A., L.D., J.W.K., H.K., Y.H. and R.H.A. provided the mice and critical comments on this study; D.Y.P. and G.Y.K. generated the figures and wrote and edited the manuscript; and G.Y.K. directed and supervised the project.

## Additional information

**Competing interests:** The authors declare no competing financial interests.

