## [Peer Review File · Nature Communications]

Reviewers' comments:

Reviewer #1 (expert in vascular cells)

Remarks to the Author:

This is a well-performed study that provides novel understanding on the mechanisms how pericyte (PC) loss destabilizes the retinal vasculature, decreasing the integrity of the blood-retinal barrier (BRB).

The authors use a number of conditional/transgenic mouse models and blocking antibodies, analyzing both postnatal and adult vasculatures. The authors identify the Ang2-FOXO1 signaling pathway as a mediator of vessel destabilization in the developing retina, in response to PC depletion. Mechanistically, impaired PDGF-B/PDGFRb signaling was associated with increased angiopoietin-2 (Ang2) and decreased Tie2 expression, leading to FOXO1 nuclear translocation and target gene expression (Fig. 3). Against the prevailing view, PC-derived Ang1 was not significant in regulation of EC-PC interactions and vessel stability in the postnatal retina (Fig. 4). Genetic inactivation of FOXO1 (Fig. 6) or Ang2 (Fig. 7), or administration of function modulating anti-Ang2 antibodies (Fig. 10) partially restored retinal vascular impairments induced by anti-PDGFRb blocking antibodies or in *Pdgfb* Δ EC mice. However, PDGF-B/PDGFRb signaling was not required for maintenance of PC coverage in adult mice. PC loss in DTA Δ PC mice increased vascular leakage in the lungs and skin, but not in the retinal vasculature, in basal conditions (Fig. S8). However, PC depletion aggravated VEGF-induced vascular permeability in the retina (Fig. 9).

Major comments:

1. p. 8. The authors state "when Tie2 was specifically depleted in the ECs at adulthood, no noticeable changes were detected in the retinal vessels compared with those of WT mice (data not shown)". Previously, *in vivo* Tie2 siRNA administered in wt mice was found to increase basal permeability in the lungs (Frye et al., 2015), therefore, the authors should provide the data for their statement that loss of Tie2 in adult mice has no effect on retinal vessels.
2. Fig. 5. Ang2+ EC area was increased approximately 30-fold upon Tie2 deletion in Tie2 Δ EC mice, but PC coverage remained unaffected. However, previous work has shown that increased Ang2 causes PC dropout in the retina (e.g. Hammes et al, 2004). – Can the authors explain why Ang2 increase did not affect PC coverage?
3. Fig. 5. The authors have previously published that global Ang1 gene deletion decreased PC coverage in the retina at P5 (Lee et al., 2013). However, in the current manuscript the authors report no change in PC coverage upon endothelial Tie2 deletion at P15, raising the question how Ang1 is exerting its effects and what is the function of Tie2 in this process. Is there an explanation for this controversy?
4. Fig. 10. The authors use the ABTAA antibody, which switches Ang2 into an Ang1-like agonist, to inhibit VEGF-induced leakage in DTA Δ PC mice. Uemura et al (2002) used recombinant Ang1 in combination with anti-PDGFRb antibody, demonstrating that increasing Tie2 activity, even in the absence of PCs, has vascular stabilizing effects in the postnatal retina. The authors should explicitly refer to this previously published data in the context of their own result.
5. Does ABTAA block the increased basal permeability in the lungs of DTA Δ PC mice?
6. Genetic deletion of FOXO1 or Ang2, or antibody blocking of Ang2 only partially rescued vascular leakage induced by PC depletion. The authors should discuss the potential reasons for this partial rescue.
7. The authors should more thoroughly introduce the previously published results demonstrating e.g. the function of PCs in postnatal retina, explored using various models (blocking antibodies, *pdgfb* gene deletions, and *pdgfbret/ret* model).

Reviewer #2 (expert in vascular cells)
Remarks to the Author:

The overall conceptual evidence connecting PDGFB signaling with the recruitment and retention of pericytes and support of vascular functions, including barrier functions, is of importance. The manuscript is based on a well-documented analysis of a series of tissue-specific knockouts and well performed in vivo models. The presented data are very impressive and are of superb quality. Overall, the data support multiple conclusions presented by the authors.

The most interesting part of the manuscript lies in the mechanistic connections between PDGFB, Tie2/Ang2 system and FOXO1 signaling and the overall role of pericytes in the regulation of these connections. In fact, I agree with the authors that the significance of this manuscript lies in the clarifications of the complexity and entanglement of the pathways connecting vascular maturation (pericytes coverage) to vascular function. The amount of experimental material is so abundant, so their interpretation often becomes confusing. The diagram at the end of the manuscript really helps to summarize the presented results and conclusions.

One of the relatively weaker points is the connection demonstrating that it is pericytes per se are responsible for Ang2 and Tie2 regulation. The authors, indeed, document the loss of pericytes and the details of the retinal phenotype associated with this loss, such as leakage, enlarged vascular structures leading to hypoxia and eventually blindness.

The connection between Ang2 and ESM1 (resulting from an enhanced transcriptional activity of FOXO1) needs to be established. Also, there is a need to show phosphorylation of FOXO1 since the nuclear localization is not very convincing.

There is a need to quantify FOXO1 activation, which is shown in Fig.3f as an example. What % of nuclei display FOXO1 presence? Changes in ESM1, VEGFR2 and Tie1 might simply coincide and occur as a result of another event. Indeed, the authors later show that Tie1 down-regulation does not seem to be critical for the phenotype.

In vitro studies show an increase in mRNA of Ang2 and ESM1 upon Tie2 silencing, however, the direct causative connection between Tie2 and pericytes coverage or PDGFB Knockout and Tie2 dysregulation need to be stronger.

Fig.3g again presents another line of evidence, which can easily be correlative. Indeed, the authors show that Tie1 was reduced in the ECs of pericyte-deficient retinal vessels; however, it is not shown that the loss of pericytes is a direct cause of Tie1 reduction. The authors show that Tie1 reduction does not cause loss vascular abnormalities, which is not exactly the same thing.

Based on S. Fig. 3, the authors conclude that inadequate pericyte coverage is the cause of changes in Ang2 expression via involvement of FOXO1. However, at least one connection in this series of events is missing. It needs to be shown directly whether and how pericyte coverage regulates FOXO1, which, in turn, regulates Ang2.

Did the authors observe the same sequence and the same changes in pericytes-depleted DTAiPC mice during retinal development? How exactly connections (coverage) with pericytes lead to the described changes? Otherwise, the presented lines of evidence might merely document an existence of parallel processes.

In fig. 5 the authors show another beautiful set of in vivo data, demonstrating the lack of involvement of pericytes-derived Ang1 in pericytes coverage, vascular morphology, and function. While the results are convincing, it is unclear why the lack of Ang1 does not disturb the balance of Ang2-Tie2. While no changes in Ang2 or Tie2 expression were observed, one can wonder whether there were any changes

in Tie2 downstream signaling at all? If it is not Ang1 that is produced by pericytes to regulate barrier and vascular function, what is? How pericytes control these processes?

Depletion of endothelial Tie2, however, led to abnormal development of retinal vasculature, characterized by leakage and proinflammatory changes.

One of the most interesting and convincing parts of this manuscript and the entire story is the "rescue" of abnormal, leaky and pericytes-deficient blood vessels (caused by inhibition of PDGFB) by endothelial-specific inactivation of FOXO1, which also diminishes Ang2 expression. Then, the authors very convincingly show that the blockade of Ang2 also rescues vascular problems, thereby solidifying this line of logic and the sequence of events.

In adult mice, it appears that depletion of PDGFB in EC or ABP5 administration does not cause the loss of pericytes demonstrating the presence of another mechanism responsible for retention of pericytes in adulthood. What these factors might be?

Likewise, deletion of pericytes in DTAiΔPC mice does not induce vascular leakage in retina or brain. It is confusing that the authors use the term "mice with detached pericytes" for two different animal models: EC-specific PDGFB KO and DTAiΔPC mice. Only the second model represents forced removal (and detachment) of pericytes. Are the phenotypes absolutely identical? I would suggest avoiding such confusions.

Next, the authors convincingly show that excess of VEGF induces the leakage and destabilization in pericytes-deficient mice. The authors show that VEGF administration in pericytes deficient mice leads to increased Ang2, VEGFR2, FOXO1 nuclear localization and vascular leakage but decreased Tie2. Is not Tie2 is already reduced as a result of pericytes loss? Please, demonstrate this point. Is Tie2 activation by ABTAA only helps in pericytes-deficient but not in normal mice treated with VEGF?

The overall gate-keeping role is centered on endothelial FOXO1, which needs to be kept suppressed to inhibit expression of destabilizing factors, such as Ang2. The authors need at least discuss what are the regulators upstream of FOXO1 and how those FOXO1 inactivators (i.e. PI3K-AKT) affect vascular leakage. In fact, several lines of evidence show that PI3K, as well as Akt inactivation causes lack of maturation and leakage, thereby supporting the authors' conclusions. The authors discuss the manuscript (ref.29), which ironically places FOXO1 upstream of Akt, whereas it is FOXO1 is a direct target of Akt signaling in most scenarios as the authors show in their diagram in Fig.10. This diagram definitely helps to deal with overwhelming amounts of data. Also, the authors have to make a final connection by discussing how exactly the pericytes keep FOXO1 inactive. There is no direct connection between pericytes and endothelial cells shown on their diagram.

Minor points:

1) In fig 1g. the authors show increased macrophage infiltration in Pdgfb iΔEC with F4/80. A significant percentage of these cells could be activated microglia as a consequence of vascular leakage (since F4/80 is used as activated microglial marker). Are the macrophages and microglia observed in the deeper regions of retina even in absence or diminished vascular plexus layer in Pdgfb iΔEC ?

2). Fig3d: authors show increased Ang2 and ESM1 in Pdgfb iΔEC . This could be due to the higher number of ECs as observed with higher CD31 staining in the thicker vessels of Pdgfb iΔEC . It might help to demonstrate this point with high magnification images showing individual ECs for comparison of Ang2 levels between WT and Pdgfb iΔEC .

3). Was the Angpt1-GFP- knock-in protein levels comparable to WT?

- 4). Can the authors show Tie2 KO at P10/12 for consistency? What is the phenotype at p30?
- 5). FOXO1 staining in fig 3f appears to be localized to the center of the cell, not just the nucleus. Higher magnification images will be more convincing.
- 6) The rescue with FOXO1 looks extremely convincing. However, the rescue with Ang2 depletion is much milder. Is it possible FOXO1 regulates transcription of other factors that are essential for vascular stability?
- 7). Why was the rescue with Ang2 blocking Ab shown on P8? How did the vessel diameter compare after the rescue?
- 8). Can developmental abnormalities in Pdgfb iΔEC or APB5 treated mice be rescued by overexpression of Tie2 or by ABTAA (P12). This might directly inhibit FOXO1 localization to nuclei and decrease Ang2 expression.
- 9). Fig 7d typo in legend: The description for fig7d in the legend labeled as (e).

Reviewer #3 (expert in diabetic retinopathy)
Remarks to the Author:

General comments

Throughout this manuscript, the authors frame the context of their data and draw conclusions which seem story- and not data driven. A series of examples is given below:

1: The abstract states: 'Impaired pericyte recruitment to the vessels showed multiple vascular hallmarks of diabetic retinopathy (DR) from BRB disruption.'

When PDGF-B was specifically depleted in ECs of Pdgfb Δ EC mice from P5, the mice showed a severe phenotype. Based on the description of the authors, this phenotype is most consistent with widespread capillary insufficiency causing a severe ischemic retinopathy with all its known sequelae, including BRB loss, inflammation, intra-retinal hemorrhage etc.

These experiments show that depletion of pericytes by interfering with the PDGF-B/PDGFRb axis causes a phenotype with similarities to diabetic retinopathy, observations which have been reported in the literature before.

In the present study, the authors have not attempted to dissect the sequence of events or causal interrelations of the various aspects of this phenotype.

Therefore, their conclusion that BRB disruption has a specific causal role in this phenotype is not supported by any observation or additional experiment and is unwarranted.

2: Page 4 states: 'These data indicate that the impaired BRB maturation caused by the initial poor pericyte coverage is permanent and could continuously worsen neuronal function in the retina, eventually leading to blindness.'

Again, it was not shown by the authors in any way that 'impaired BRB maturation' is specifically caused by pericyte depletion, nor was it shown that the deficient BRB in the severely diseased retina at P30 has a specific causal role in this phenotype. These experiments only show that the retina in these experiments was beyond repair after induction of the severe phenotype at P12 and persisting PDGF depletion.

3: Page 6 states: 'Thus, adequate pericyte coverage is required to strengthen BRB integrity against further vascular impairments, including severe ischemia, inflammation, and oxidative stresses.'

The authors have indeed shown that the OIR phenotype is worse in the context of PDGF depletion, but evidence that 'strengthening the BRB' would have anything to do with that effect was not provided.

4: Page 7 states: 'Thus, the ECs became unstable with inadequate pericyte coverage, and cleavage of Tie1 led to suppression of Tie2 availability for inhibition of FOXO1 activity of Ang2 transcription^{32,33}. In fact, Chip-PCR sequencing analysis revealed that FOXO1 binding was enriched in promoter and enhancer regions of Ang2 (Supplementary Fig. 3d,e), indicating that FOXO1 directly regulates Ang2 expression in the destabilized ECs caused by inadequate pericyte coverage.'

Although the data presented in this paragraph are interesting and hypothesis-generating, they do not support these conclusions. In particular the direct role of pericyte coverage in regulating Ang2 expression was not shown. In fact, HUVEC in vitro have no pericyte coverage either but were used as a model anyway.

2: The abstract states: 'Accordingly, either blocking Ang2 or activating Tie2 greatly ameliorated progression of BRB breakdown, providing beneficial effects in delaying DR progression by preventing BRB disintegration

In the adult mice with reduced pericyte coverage caused by DTAi Δ PC mice, the BRB was not affected, arguing against a major maintenance role of pericytes in the BRB in mice. The authors show data that this led to increased BRB loss from a single injection of exogenous VEGF-A and a retinopathy similar to the previously reported effects of repeated exogenous VEGF-A injections in monkeys. The observed effects of VEGF-A in mice in this study could be blocked by Tie2 activation. These observations are all very interesting and mostly novel. However, the effect of Ang2 inhibition mentioned in the abstract in this context cannot be retrieved in the Results section. Furthermore, what is meant by 'progression of BRB breakdown', as the experiment was based on one time point only?

More importantly, the extrapolation to DR progression by the authors and the claim that DR progression could be delayed by preventing BRB disintegration are farfetched. DR progression in mice is not identical to increasing BRB loss, but is characterized by vasoregression, the loss of capillaries

Answers to the editor's and reviewers' comments

We deeply appreciate the editor and reviewers for their thoughtful, critical and constructive comments, which have undoubtedly provided us with valuable opportunities to improve our work. We have performed additional experiments and revised the manuscript to address the issues raised by the reviewers.

Reviewer #1

This is a well-performed study that provides novel understanding on the mechanisms how pericyte (PC) loss destabilizes the retinal vasculature, decreasing the integrity of the blood-retinal barrier (BRB). The authors use a number of conditional/transgenic mouse models and blocking antibodies, analyzing both postnatal and adult vasculatures. The authors identify the Ang2-FOXO1 signaling pathway as a mediator of vessel destabilization in the developing retina, in response to PC depletion. Mechanistically, impaired PDGF-B/PDGFR β signaling was associated with increased angiopoietin-2 (Ang2) and decreased Tie2 expression, leading to FOXO1 nuclear translocation and target gene expression (Fig. 3). Against the prevailing view, PC-derived Ang1 was not significant in regulation of EC-PC interactions and vessel stability in the postnatal retina (Fig. 4). Genetic inactivation of FOXO1 (Fig. 6) or Ang2 (Fig. 7), or administration of function modulating anti-Ang2 antibodies (Fig. 10) partially restored retinal vascular impairments induced by anti-PDGFRb blocking antibodies or in *Pdgfb* ^{Δ EC} mice. However, PDGF-B/PDGFR β signaling was not required for maintenance of PC coverage in adult mice. PC loss in *DTA* ^{Δ PC} mice increased vascular leakage in the lungs and skin, but not in the retinal vasculature, in basal conditions (Fig. S8). However, PC depletion aggravated VEGF-induced vascular permeability in the retina (Fig. 9).

We appreciate these favorable and encouraging comments.

Comment 1. p.8. The authors state “when Tie2 was specifically depleted in the ECs at adulthood, no noticeable changes were detected in the retinal vessels compared with those of WT mice (data not shown)”. Previously, in vivo Tie2 siRNA administered in WT mice was found to increase basal permeability in the lungs (Frye et al., 2015), therefore, the authors should provide the data for their statement that loss of Tie2 in adult mice has no effect on retinal vessels.

We appreciate this constructive comment. We included the data and their description regarding the retinal vascular phenotype of Tie2 deletion in ECs during adulthood into the revised manuscript (page 8, Supplementary Fig. 10). Dependency on Tie2 for vascular stability and integration could be an organ-specific during adulthood. In fact, lung vessels continue to require Tie2, while the retinal vessels no longer require Tie2 during adulthood. Accordingly, while lung vessels have abundant expression of Tie2, retinal vessels have very low levels of Tie2 during adulthood. We added this statement into the revised manuscript (page 8).

However, when Tie2 was specifically depleted in the ECs at adulthood, no noticeable changes including vascular leakage were detected in the retinal vessels compared with those of WT mice (Supplementary Fig. 10).

Comment 2. Fig. 5. Ang2+ EC area was increased approximately 30-fold upon Tie2 deletion in Tie2^{iΔEC} mice, but PC coverage remained unaffected. However, previous work has shown that increased Ang2 causes PC dropout in the retina (e.g. Hammes et al, 2004). – Can the authors explain why Ang2 increase did not affect PC coverage?

This is a critical point that needs to be more addressed. In fact, we also paid attention to this unexpected finding because Ang2 is known to be a positive pericyte dropout factor (Hammes et al., 2004). However, the roles of Ang2 in the PC dropout may also be context dependent. It could be dependent on status of vessels (growing or stabilized), source (endogenous or exogenous), levels of Tie2, Tie1, Ang1, VEGF and integrin, microenvironment (hypoxia, inflammation), organ, and experimental setting. Similar explanation and interpretation have been addressed and highlighted regarding the roles of Ang2 in inflammation and acute endotoxemia conditions (Kim et al., 2016; Korhonen et al., 2016). In this study, the up-regulated Ang2 may have multiple actions, which counter the pericyte dropout effects of Ang2. We modified two sentences in the revised manuscript as follows (page 14):

In fact, Ang2 *per se* is a positive factor for pericyte dropout in animal models of DR (Hammes et al., 2004; Pfister et al., 2010), although its role in pericyte dropout could be overridden in a context-dependent manner as was the case in the Tie2^{iΔEC} mice in this study (Fig. 5f).

Comment 3. Fig. 5. The authors have previously published that global Ang1 gene deletion decreased PC coverage in the retina at P5 (Lee et al., 2013). However, in the current manuscript the authors report no change in PC coverage upon endothelial Tie2 deletion at P15, raising the question how Ang1 is exerting its effects and what is the function of Tie2 in this process. Is there an explanation for this controversy?

We appreciate this critical comment, which requires a reasonable explanation with our unpublished preliminary data. As we have shown in the original manuscript (Fig. 4), the main source of Ang1 is the retinal neurons located in the ganglion cell layer and inner nuclear layer. The retinal neuronal derived-Ang1 could be involved in retinal angiogenesis and pericyte recruitment through Tie2 and integrin signaling during the postnatal development. In comparison, the vascular phenotype of endothelial Tie2 deletion in growing retinal vessels showed more severe vascular impairment, but did not affect pericyte coverage (Fig. 5). Blood-retinal barrier is composed of ECs, pericytes, and astrocytes. These cells functionally and physically interact closely with each other, as like in blood-brain barrier where the pericytes change the polarization of adjacent astrocytes (Armulik et al., 2010). Therefore, it could be possible that neuron-derived Ang1 could affect the ECs via both Tie2 and

integrin signaling pathways or astrocytes via integrin signaling pathways, and indirectly affect recruitment or attachment of pericytes, which need to be validated by a separate, future study.

We modified these sentences in the revised manuscript as follows (page 12):

In contrast to the prevailing concept, we provide compelling evidence that pericytes that surround retinal vessels do not express Ang1. Despite the lack of Ang1 expression by pericytes, globally Ang1-depleted neonatal mice show reduced vascular growth and pericyte coverage in the retina (Lee et al., 2013). These collectively imply that Ang1 may have derived from adjacent neurons or hematopoietic cells including platelets, which subsequently activate Tie2 or integrins for its roles in formation and maturation of the inner part of the BRB.

Comment 4. Fig. 10. The authors use the ABTAA antibody, which switches Ang2 into an Ang1-like agonist, to inhibit VEGF-induced leakage in DTA^{iΔPC} mice. Uemura et al (2002) used recombinant Ang1 in combination with anti-PDGFR β antibody, demonstrating that increasing Tie2 activity, even in the absence of PCs, has vascular stabilizing effects in the postnatal retina. The authors should explicitly refer to this previously published data in the context of their own result.

We appreciate this constructive comment. We referred to a previous paper (Uemura et al., 2002) and modified the description accordingly in the revised manuscript as follows (page 11):

Compared with Fc, ABTAA reduced vascular leakage and macrophage infiltration without restoration of pericyte coverage onto the vessels (**Fig. 10b, c**), which is similar to the findings reported previously (Uemura et al., 2002). Thus, Tie2 activation could ameliorate BRB disintegration against VEGF-A stimuli in the absence of pericytes, preventing retinal damages during DR progression.

Comment 5. Does ABTAA block the increased basal permeability in the lungs of DTA^{iΔPC} mice?

In response to the reviewer's question, we performed an additional experiment to examine the effects of ABTAA on increased basal permeability in the lungs of DTA^{iΔPC} mice (below). As expected, ABTAA significantly reduced the permeability in the lung. We show these results only for the reviewer because we think that this is beyond the scope of this study. We are currently analyzing various organs including the lungs in the DTA^{iΔPC} mice and are trying to figure out how pericytes differently regulate the vascular integrity in an organ-specific manner and the effect of ABTAA on vascular leakage as a separate study. We would be appreciated if the reviewer considers this situation.

(a) Diagram depicting the experiment schedule for selective loss of pericytes in 8-week-old DTA^{iΔPC} mice and administration of ABTAA (20 mg/kg). Analyses of vascular leakage of the lung was performed by Evans blue (EB) injection. (b, c) Images and comparisons of the lung of Fc-treated (Fc, n = 3) and ABTAA-treated (AB, n = 3) DTA^{iΔPC} mice after EB injection are shown. Error bars represent mean \pm s.d. *P < 0.05 versus Fc by Mann-Whitney U test.

Comment 6. Genetic deletion of FOXO1 or Ang2, or antibody blocking of Ang2 only partially rescued vascular leakage induced by PC depletion. The authors should discuss the potential reasons for this partial rescue.

We appreciate this comment. There could be several reasons for the partial rescue by genetic deletion of FOXO1 or Ang2 or Ang2 blocking antibody. First of all, there could be unidentified molecular actions or pathways that are responsible for the pericyte depletion-induced vascular leakage but independent of action of FOXO1 or Ang2. Second, the deletion efficiency of FOXO1 or Ang2 by tamoxifen may not have been 100% to fully inactivate the action of FOXO1 or Ang2; similarly, the delivery and actions of Ang2-blocking antibody would not have applied to all existing Ang2. Third, the anti-vascular leakage effect of FOXO1 deletion was stronger than that of Ang2 deletion or blockade. As a transcriptional factor, FOXO1 up-regulates expressions of several key genes including ESM1, VEGFR2 in addition to Ang2 (Daly et al., 2004; Dharaneeswaran et al., 2014; Potente et al., 2005), which are involved in vascular destabilization and leakage. To address this comment, we additionally included following description into the revised Discussion (page 14):

In the present study, our approaches with genetic deletion of Ang2 or FOXO1, or administration of Ang2 blocking antibody highlight the critical roles of both Ang2 and FOXO1 as negative regulators of EC stability in the absence of pericytes. However, neither genetic deletion nor antibody administration fully mitigated the vascular leakage caused by inadequate pericyte coverage. There could be several reasons for this partial rescue. First, there could be unidentified molecular actions or pathways that are responsible for the pericyte depletion-induced vascular leakage but independent of the

action of FOXO1 or Ang2. Second, the deletion efficiency may not have been 100% to fully inactivate the actions of FOXO1 or Ang2; similarly, Ang2 blocking antibody would not have perfectly acted on all existing Ang2 due to issues with drug delivery and binding efficiencies.

Comment 7. The authors should more thoroughly introduce the previously published results demonstrating e.g. the function of PCs in postnatal retina, explored using various models (blocking antibodies, pdgfb gene deletions, and pdgfb^{ret/ret} model).

We appreciate this constructive comment. As the reviewer recommended, we introduced the previously published results regarding the function of PCs more thoroughly as follows (page 3):

Previous landmark studies revealed that proper pericyte recruitment to growing retinal and brain vessels through controlled platelet-derived growth factor (PDGF)-B/PDGF receptor beta (PDGFR β) signaling is critical for adequate formation of the BRB and BBB. Genetic deletion of PDGF-B in ECs or its retention motif that is responsible for the retention of PDGF-B within the pericellular space in the growing retinal vessels exhibited severe vascular impairments including vascular engorgement and leakage together with improper pericyte recruitment and organization during the retinal vascular development (Enge et al., 2002; Lindblom et al., 2003), which are similar to the events induced by a PDGFR β blocking antibody (Uemura et al., 2002). These studies indicated that PDGF-B secreted from and properly retained in retinal vascular ECs recruit PDGFR β ⁺ pericytes to facilitate vascular growth and stabilization during vascular development.

***References for responses to comments of reviewer 1**

Armulik, A., Genove, G., Mae, M., Nisancioglu, M.H., Wallgard, E., Niaudet, C., He, L., Norlin, J., Lindblom, P., Strittmatter, K., et al. (2010). Pericytes regulate the blood-brain barrier. *Nature* 468, 557-561.

Daly, C., Wong, V., Burova, E., Wei, Y., Zabski, S., Griffiths, J., Lai, K.M., Lin, H.C., Ioffe, E., Yancopoulos, G.D., et al. (2004). Angiopoietin-1 modulates endothelial cell function and gene expression via the transcription factor FKHR (FOXO1). *Genes Dev* 18, 1060-1071.

Dharaneeswaran, H., Abid, M.R., Yuan, L., Dupuis, D., Beeler, D., Spokes, K.C., Janes, L., Sciuto, T., Kang, P.M., Jaminet, S.C., et al. (2014). FOXO1-mediated activation of Akt plays a critical role in vascular homeostasis. *Circ Res* 115, 238-251.

Enge, M., Bjarnegard, M., Gerhardt, H., Gustafsson, E., Kalen, M., Asker, N., Hammes, H.P., Shani, M., Fassler, R., and Betsholtz, C. (2002). Endothelium-specific platelet-derived growth factor-B ablation mimics diabetic retinopathy. *EMBO J* 21, 4307-4316.

Hammes, H.P., Lin, J., Wagner, P., Feng, Y., Vom Hagen, F., Krzizok, T., Renner, O., Breier, G., Brownlee, M., and Deutsch, U. (2004). Angiopoietin-2 causes pericyte dropout in the normal retina: evidence for involvement in diabetic retinopathy. *Diabetes* 53, 1104-

1110

Kim, M., Allen, B., Korhonen, E.A., Nitschke, M., Yang, H.W., Baluk, P., Saharinen, P., Alitalo, K., Daly, C., Thurston, G., et al. (2016). Opposing actions of angiotensin-2 on Tie2 signaling and FOXO1 activation. *J Clin Invest* 126, 3511-3525.

Korhonen, E.A., Lampinen, A., Giri, H., Anisimov, A., Kim, M., Allen, B., Fang, S., D'Amico, G., Sipila, T.J., Lohela, M., et al. (2016). Tie1 controls angiotensin function in vascular remodeling and inflammation. *J Clin Invest* 126, 3495-3510.

Lee, J., Kim, K.E., Choi, D.K., Jang, J.Y., Jung, J.J., Kiyonari, H., Shioi, G., Chang, W., Suda, T., Mochizuki, N., et al. (2013). Angiotensin-1 guides directional angiogenesis through integrin α v β 5 signaling for recovery of ischemic retinopathy. *Sci Transl Med* 5, 203ra127.

Lindblom, P., Gerhardt, H., Liebner, S., Abramsson, A., Enge, M., Hellstrom, M., Backstrom, G., Fredriksson, S., Landegren, U., Nystrom, H.C., et al. (2003). Endothelial PDGF-B retention is required for proper investment of pericytes in the microvessel wall. *Genes Dev* 17, 1835-1840.

Pfister, F., Wang, Y., Schreiter, K., vom Hagen, F., Altvater, K., Hoffmann, S., Deutsch, U., Hammes, H.P., and Feng, Y. (2010). Retinal overexpression of angiotensin-2 mimics diabetic retinopathy and enhances vascular damages in hyperglycemia. *Acta Diabetol* 47, 59-64.

Potente, M., Urbich, C., Sasaki, K., Hofmann, W.K., Heeschen, C., Aicher, A., Kollipara, R., DePinho, R.A., Zeiher, A.M., and Dimmeler, S. (2005). Involvement of Foxo transcription factors in angiogenesis and postnatal neovascularization. *J Clin Invest* 115, 2382-2392.

Uemura, A., Ogawa, M., Hirashima, M., Fujiwara, T., Koyama, S., Takagi, H., Honda, Y., Wiegand, S.J., Yancopoulos, G.D., and Nishikawa, S. (2002). Recombinant angiotensin-1 restores higher-order architecture of growing blood vessels in mice in the absence of mural cells. *J Clin Invest* 110, 1619-1628.

Reviewer #2

The overall conceptual evidence connecting PDGFB signaling with the recruitment and retention of pericytes and support of vascular functions, including barrier functions, is of importance. The manuscript is based on a well-documented analysis of a series of tissue-specific knockouts and well performed in vivo models. The presented data are very impressive and are of superb quality. Overall, the data support multiple conclusions presented by the authors. The most interesting part of the manuscript lies in the mechanistic connections between PDGFB, Tie2/Ang2 system and FOXO1 signaling and the overall role of pericytes in the regulation of these connections. In fact, I agree with the authors that the significance of this manuscript lies in the clarifications of the complexity and entanglement of the pathways connecting vascular maturation (pericytes coverage) to vascular function. The amount of experimental material is so abundant, so their interpretation often becomes confusing. The diagram at the end of the manuscript really helps to summarize the presented results and conclusions.

We appreciate these favorable and encouraging comments.

Major comment 1. One of the relatively weaker points is the connection demonstrating that it is pericytes per se are responsible for Ang2 and Tie2 regulation. The authors, indeed, document the loss of pericytes and the details of the retinal phenotype associated with this loss, such as leakage, enlarged vascular structures leading to hypoxia and eventually blindness. The connection between Ang2 and ESM1 (resulting from an enhanced transcriptional activity of FOXO1) needs to be established. Also, there is a need to show phosphorylation of FOXO1 since the nuclear localization is not very convincing. There is a need to quantify FOXO1 activation, which is shown in Fig.3f as an example. What % of nuclei display FOXO1 presence? Changes in ESM1, VEGFR2 and Tie1 might simply coincide and occur as a result of another event. Indeed, the authors later show that Tie1 down-regulation does not seem to be critical for the phenotype.

We appreciate this constructive comment. Previous reports (Daly et al., 2004; Potente et al., 2005) have demonstrated that expressions of Ang2 and ESM1 are positively regulated by FOXO1. To confirm these findings, we performed ChIP-qPCR analysis for FOXO1 binding to the enhancers and promoter of Ang2, and included the results as revised Supplementary Fig. 6 and their description in the original text (page 7).

We unfortunately failed to detect phospho-FOXO1 in the retinal vessels using commercially available antibodies. Instead, to quantify FOXO1 activation in the retinal vessels, we regarded the round and condensed signal of FOXO1 immunofluorescence staining in the retinal vessels as a nuclear-localized activated form of FOXO1, in accordance to a recently published report (Kim et al., 2016). Then, area of the nuclear-localized FOXO1 per CD31⁺ EC area was calculated and presented as “nuclear FOXO1/ECs (%). We would like to let the reviewer know that

we were able to distinguish high signal density of FOXO1 in the nuclei from low signal density of FOXO1 in the cytoplasm, which may convincingly reflect the FOXO1 activity. We included these descriptions into the revised Method (page 18). We also included the reanalyzed data of “nuclear FOXO1/ECs (%)” as Fig. 3, 6 and Supplementary Fig. 3, 5, 15, 16 into the revised manuscript.

We regarded the round and condensed signal of FOXO1 immunofluorescence staining in the retinal vessels as a nuclear-localized activated form of FOXO1, in accordance to a recently published report (Kim et al., 2016). Then, area of the nuclear-localized FOXO1 per CD31⁺ EC area was calculated using same threshold values using ImageJ software and presented as “nuclear FOXO1/ECs (%)”. Similarly, to quantify Ang2 or ESM1 expression, their stained area per CD31⁺ EC area was calculated and presented as “Ang2/ECs (%)” or “ESM1/ECs (%)”. If needed, their values were normalized by control and presented as fold change.

As the reviewer pointed out, the changes of ESM1, VEGFR2 and Tie1 could simply be coincident. However, our extensive analyses demonstrated the cause-sequential relationship of inadequate pericyte coverage-Tie2/Tie1-PI3K-Akt signaling-FOXO1-mediated transcriptional changes of Ang2, ESM1 and VEGFR2 in a positive feedback manner, as we summarized in original Fig. 10. This mechanistic insight is consistent with two recent reports (Kim et al., 2016; Korhonen et al., 2016).

We presented the reduced level of Tie1 in the pericyte-deficient retinal vessels, however, we did not study the effects of Tie1 down-regulation on the phenotype. Please consider this caveat.

Major comment 2. In vitro studies show an increase in mRNA of Ang2 and ESM1 upon Tie2 silencing, however, the direct causative connection between Tie2 and pericytes coverage or PDRFB Knockout and Tie2 dysregulation need to be stronger. Fig.3g again presents another line of evidence, which can easily be correlative. Indeed, the authors show that Tie1 was reduced in the ECs of pericyte-deficient retinal vessels; however, it is not shown that the loss of pericytes is a direct cause of Tie1 reduction. The authors show that Tie1 reduction does not cause loss vascular abnormalities, which is not exactly the same thing.

This is a valid point that needs to be cautiously interpreted.

In original Supplementary Fig. 3c, we validated that up-regulation of Ang2 and ESM1 upon Tie2 depletion can be mediated by FOXO1 by analyzing the mRNA levels after transfecting HUVECs with siFOXO1 and/or siTie2.

The mechanism by which pericyte coverage regulates Tie2 can be explained in two ways. First, Tie2 shedding (cleavage of Tie2 extracellular domain) in ECs could have occurred immediately by pericyte loss, because nascent ECs are directly exposed to a bunch of activated proteases and growth factors such as VEGF-A. In fact, VEGF-A is known to induce Tie2 shedding via a PI3K/Akt dependent pathway to modulate

Tie2 signaling (Findley et al., 2007). Second, as we described in the original manuscript (page 13), inflammation-induced cleavage of Tie1 extracellular domain can lead to reduced availability and signaling of Tie2, as two recent papers (Kim et al., 2016; Korhonen et al., 2016) have demonstrated. Thus, inflammation induced by the BRB disruption from the pericyte loss could also reduce functional Tie2 level.

In the original manuscript, we did not analyze the effect of Tie1 reduction in retinal vascular development and BRB function. In fact, a previous study (Savant et al., 2015) revealed that EC-specific Tie1 deletion reduces radial growth and vascular density in growing retinal vessels of postnatal mice.

We modified these statements in the revised manuscript (page 7).

Furthermore, Tie1 was reduced in the ECs of pericyte-detached retinal vessels, while soluble Tie1 (cleaved ecto-domain) was increased in the vitreous fluid of *Pdgfb*^{ΔEC} mice (Fig. 3h,i); these are similar to those which occur to ECs upon acute inflammation (Kim et al., 2016; Korhonen et al., 2016). These data suggested that the ECs became unstable due to inadequate pericyte coverage, and that cleavage of Tie1 led to the suppression of Tie2 availability, which is required to inhibit Ang2 transcription by FOXO1.

We also included these statements into the revised manuscript (page 13).

The mechanism by which pericyte coverage regulates Tie2 can be explained in two ways. First, Tie2 shedding (cleavage of Tie2 extracellular domain) in ECs could have occurred immediately after pericyte loss, because ECs are directly exposed to a bunch of activated proteases and growth factors such as VEGF-A. In fact, VEGF-A is known to induce Tie2 shedding via a phosphatidylinositol 3 (PI3) kinase/Akt-dependent pathway to modulate Tie2 signaling (Findley et al., 2007). In addition, recent elegant studies (Kim et al., 2016; Korhonen et al., 2016; Singh et al., 2012) have revealed that inflammation-induced cleavage of the Tie1 extracellular domain leads to reduced availability and signaling of Tie2, which in turn further degrades vascular stability through FOXO1-mediated up-regulation of Ang2 in the ECs of tracheal vessels. Thus, in agreement with these studies, inflammation induced by the BRB disruption after pericyte loss could also reduce functional Tie2 level and stimulate FOXO1 activation.

Major comment 3. Based on S.Fig.3, the authors conclude that inadequate pericyte coverage is the cause of changes in Ang2 expression via involvement of FOXO1. However, at least one connection in this series of events is missing. It needs to be shown directly whether and how pericyte coverage regulates FOXO1, which, in turn, regulates Ang2.

Did the authors observe the same sequence and the same changes in pericytes-depleted *DTA*^{ΔPC} mice during retinal development? How exactly connections (coverage) with pericytes lead to the described changes? Otherwise, the presented lines of evidence might merely document an existence of parallel processes.

We agree that this is a very important issue that needs to be addressed further because it has been mentioned several times by the reviewer. In fact, we already

showed a possible connection of “inadequate pericyte coverage (by treatment of PDGFR β blocking antibody, APB5)/up-regulation and nuclear localization of FOXO1/Ang2 up-regulation in the ECs” as original Fig. 6. We also validated this possible connection using *Foxo1*^{iAEC} mice in original Fig. 6.

Nevertheless, to further address this connection, we performed an additional experiment (Supplementary Fig. 5) by employing an *ex vivo* aortic ring assay (Baker et al., 2011) using the aortas of WT and *Pdgfb*^{iAEC} mice. With this assay, we were able to detach PDGFR β ⁺ or NG2⁺ pericytes from ECs in sprouting vessels of the aorta of *Pdgfb*^{iAEC} mice, while keeping the pericyte-EC interaction intact in those of the aorta from WT mice. Moreover, this assay excludes the possible effects from neurohormonal cells or inflammatory cells, which occur *in vivo*. The pericyte-detached ECs exhibited high levels and nuclear localization of FOXO1 and high level of Ang2, and intriguingly the detached pericytes also exhibited high levels and nuclear localization of FOXO1 (Supplementary Fig. 5). In contrast, no major changes in FOXO1 and Ang2 were detected in the sprouting vessels with intact pericytes in the aorta of WT mice (Supplementary Fig. 5).

As the reviewer recommended, we also performed an additional experiment with DTA^{iAPC} mice to investigate this connection (Supplementary Fig. 15). Pericyte detachment in the growing retinal vessels of DTA^{iAPC} mice not only up-regulated FOXO1 but also induced nuclear localization of FOXO1 as well as up-regulating Ang2 in the ECs. Thus, pericyte detachment appears to up-regulate and activate FOXO1 directly in ECs of the retinal vessels, which leads to stimulation of Ang2 expression in ECs of the vessels.

We included these results and related description into the revised manuscript (page 7, 10)

Next, to reveal whether and how pericyte coverage regulates FOXO1 and Ang2, we employed an *ex vivo* aortic ring assay (Baker et al., 2011) using the aortas of WT and *Pdgfb*^{iAEC} mice, where the mice were pre-administrated with tamoxifen before aorta harvest (**Supplementary Fig. 5a**). During *ex vivo* culture, the PDGFR β ⁺ pericytes were detached from the sprouts, which were reduced in number and length themselves, of the aorta of *Pdgfb*^{iAEC} mice, and the pericyte-detached ECs had higher levels of Ang2 and FOXO1 as well as nuclear localization of FOXO1 (**Supplementary Fig. 5b-e**). In contrast, no major changes in FOXO1 or Ang2 were detected in the ECs of sprouts with intact pericytes in the aorta of WT mice (**Supplementary Fig. 5d,e**). Thus, pericyte detachment appears to directly up-regulate and activate FOXO1 in ECs, which leads to stimulation of Ang2 expression in ECs. In fact, Chip-PCR sequencing analysis revealed that FOXO1 binding was enriched in the promoter and enhancer regions of Ang2 (**Supplementary Fig. 6a,b**), indicating that FOXO1 directly regulates Ang2 expression in the destabilized ECs caused by inadequate pericyte coverage (page 7).

On the other hand, impaired retinal vascular maturation together with BRB disruption, which were similar to the findings of *Pdgfb*^{iAEC} mice, were observed in DTA^{iAPC} mice during postnatal development (**Supplementary Fig. 15a-c**). These findings indicate that

the regulation of pericytes on BRB could be dependent on stability of the vessels, and a sudden pericyte loss from the stabilized retinal vessels is insufficient to induce a significant alteration in the integrity of retinal vessels and BRB, and additional insults may be required to mimic the hallmarks of DR (page 10).

Major Comment 4. In fig. 5 the authors show another beautiful set of in vivo data, demonstrating the lack of involvement of pericytes-derived Ang1 in pericytes coverage, vascular morphology, and function. While the results are convincing, it is unclear why the lack of Ang1 does not disturb the balance of Ang2-Tie2. While no changes in Ang2 or Tie2 expression were observed, one can wonder whether there were any changes in Tie2 downstream signaling at all? If it is not Ang1 that is produced by pericytes to regulate barrier and vascular function, what is? How pericytes control these processes? Depletion of endothelial Tie2, however, led to abnormal development of retinal vasculature, characterized by leakage and proinflammatory changes.

We appreciate this critical comment, which requires a reasonable explanation with our unpublished preliminary data. As we shown in the original manuscript (Fig. 4), the main source of Ang1 is the retinal neurons located in the ganglion cell layer and inner nuclear layer. The retinal neuronal derived-Ang1 could be involved in retinal angiogenesis and pericyte recruitment through Tie2 and integrin signaling during the postnatal development. In comparison, the vascular phenotype of endothelial Tie2 deletion in growing retinal vessels showed more severe vascular impairment (Fig. 5). Thus, Tie2 can be activated by Ang1 derived from sources other than pericytes. More importantly, it is well known that clustering of Tie2 is critical for its activation . The clustering of Tie2 can be strongly induced by a shear stress, independent of Ang1. Thus, the abnormal vascular phenotypes in the endothelial Tie2 deletion can be mediated by the absence of Tie2 clustering induced by a blood-flow-mediated shear stress.

We modified these sentences in our revised manuscript as follows (page 12);

In contrast to the prevailing concept, we provide compelling evidence that pericytes that surround retinal vessels do not express Ang1. Despite the lack of Ang1 expression by pericytes, globally Ang1-depleted neonatal mice show reduced vascular growth and pericyte coverage in the retina. These collectively imply that Ang1 may have derived from adjacent neurons or hematopoietic cells including platelets, which subsequently activate Tie2 or integrins for its roles in formation and maturation of the inner part of the BRB. In the context of Tie2, we showed that, unlike Ang1, endothelial Tie2 plays an indispensable role in vascular development and BRB maturation. It is well known that clustering of Tie2, usually induced by Ang1, is critical for its activation (Augustin et al., 2009; Kim et al., 2005; Koh, 2013). However, the clustering of Tie2 can also be strongly induced by shear stress, which is an independent mode of action to Ang1-induced activation. Thus, the abnormal vascular phenotypes after endothelial Tie2 deletion can be due to the absence of Tie2 clustering induced by a blood flow-mediated shear stress.

Major comment 5. One of the most interesting and convincing parts of this

manuscript and the entire story is the “rescue” of abnormal, leaky and pericytes-deficient blood vessels (caused by inhibition of PDGFB) by endothelial-specific inactivation of FOXO1, which also diminishes Ang2 expression. Then, the authors very convincingly show that the blockade of Ang2 also rescues vascular problems, thereby solidifying this line of logic and the sequence of events. In adult mice, it appears that depletion of PDGFB in EC or APB5 administration does not cause the loss of pericytes demonstrating the presence of another mechanism responsible for retention of pericytes in adulthood. What these factors might be?

This is an important question but we do not have a definitive answer. Physically stable interactions among the surrounding cells and their basement membrane and extracellular matrix could be the major components responsible for the pericyte retention in adulthood.

Major comment 6. Likewise, deletion of pericytes in $DTA^{\Delta PC}$ mice does not induce vascular leakage in retina or brain. It is confusing that the authors use the term “mice with detached pericytes” for two different animal models: EC-specific PDGFB KO and $DTA^{\Delta PC}$ mice. Only the second model represents forced removal (and detachment) of pericytes. Are the phenotypes absolutely identical? I would suggest avoiding such confusions.

We agree with this comment. As the reviewer recommended, in revised manuscript, we did not use the term ‘detachment’ or ‘depletion’ in the $Pdgfb^{\Delta EC}$ mice and used the terms ‘inadequate pericyte coverage’ or ‘inhibition of pericyte recruitment’ instead (page 7, 8).

Similar up-regulation in the expressions of ESM1 and VEGFR2, which are highly enriched in the pericyte-uncovered tip ECs, and distinct nuclear localization of FOXO1 were found in the ECs of pericyte-uncovered retinal vessels (**Fig. 3f-i**).

To address how FOXO1-induced Ang2 up-regulation contributes to vascular impairments in pericyte-uncovered retinal vessels, we crossed the $Foxo1^{flox/flox}$ mouse with the $VE-cadherin-Cre-ER^{T2}$ mouse and generated the $Foxo1^{\Delta EC}$ mouse (**Fig. 6a**). Then, inadequate pericyte coverage in the $Foxo1^{\Delta EC}$ mouse was generated by intra-peritoneal administration of the PDGFR β + blocking antibody APB5 at P1 (**Fig. 6a**).

Major comment 7. Next, the authors convincingly show that excess of VEGF induces the leakage and destabilization in pericytes-deficient mice. The authors show that VEGF administration in pericytes deficient mice leads to increased Ang2, VEGFR2, FOXO1 nuclear localization and vascular leakage but decreased Tie2. Is not Tie2 is already reduced as a result of pericytes loss? Please, demonstrate this point.

This is a valid point that needs to be explained in more detail. In contrast to growing vessels, there was no change in Tie2 by pericyte depletion in adult stable retinal vessels (Supplementary Fig. 15, 16). Thus, different changes of Tie2 upon pericyte

depletion seem to be dependent on stability of the vessels.

As we described in original manuscript as a schematic diagram (Fig. 10d), in adult stable retinal vessels, several key factors including Tie2/Tie1- and shear stress-mediated activation of PI3K-Akt signaling play key roles in the maintenance of BRB integrity. Tie2 could be dysregulated by external stimuli, such as VEGF-A, or inflammatory microenvironment upon pericyte depletion (Findley et al., 2007; Kim et al., 2016; Korhonen et al., 2016; Singh et al., 2012) as addressed in our response to comment 2. Thus, in adult vessels, Tie2 can be attenuated by pericyte loss, when exposed to external noxious stimuli, such as VEGF-A.

We included these sentences in our revised manuscript as follows (page 10, 13):

On the other hand, impaired retinal vascular maturation together with BRB disruption, which were similar to the findings of *Pdgfb*^{iAEC} mice, were observed in *DTA*^{iAPC} mice during postnatal development (**Supplementary Fig. 15a-c**). These findings indicate that the regulation of pericytes on BRB could be dependent on stability of the vessels, and a sudden pericyte loss from the stabilized retinal vessels is insufficient to induce a significant alteration in the integrity of retinal vessels and BRB, and additional insults may be required to mimic the hallmarks of DR (page 10).

The mechanism by which pericyte coverage regulates Tie2 can be explained in two ways. First, Tie2 shedding (cleavage of Tie2 extracellular domain) in ECs could have occurred immediately after pericyte loss, because ECs are directly exposed to a bunch of activated proteases and growth factors such as VEGF-A. In fact, VEGF-A is known to induce Tie2 shedding via a phosphatidylinositol 3 (PI3) kinase/Akt-dependent pathway to modulate Tie2 signaling (Findley et al., 2007). In addition, recent elegant studies (Kim et al., 2016; Korhonen et al., 2016; Singh et al., 2012) have revealed that inflammation-induced cleavage of the Tie1 extracellular domain leads to reduced availability and signaling of Tie2, which in turn further degrades vascular stability through FOXO1-mediated up-regulation of Ang2 in the ECs of tracheal vessels. Thus, in agreement with these studies, inflammation induced by the BRB disruption after pericyte loss could also reduce functional Tie2 level and stimulate FOXO1 activation. **Nevertheless, unlike FOXO1 and Ang2, the change in Tie2 upon pericyte depletion was different between growing retinal vessels and stable adult retinal vessels, implying that regulation of Tie2 by pericytes could also be dependent on vessel stability as well as its microenvironment, such as inflammation and hypoxia (page 13).**

Major comment 8. Is Tie2 activation by ABTAA only helps in pericytes-deficient but not in normal mice treated with VEGF?

As shown in Fig. 9, intravitreal administration of VEGF-A to WT normal mice also significantly increased vascular leakage, although the leakage was much less than that of *DTA*^{iAPC} mice. However, as we have shown below, ABTAA treatment did not reduce the vascular leakages in WT mice induced by intravitreal VEGF administration. We speculate that this could be attributed to the fact that Ang2 expression was not induced in the retinal vessels by the VEGF treatment in WT mice (Fig. 9b-c), as Tie2 agonistic action of ABTAA works only in the presence of Ang2

(Han et al., 2016). We show this result only for the reviewer.

(a) Diagram depicting the experiment schedule for tamoxifen injections, intra-vitreous administration of VEGF-A (1 μ g), and daily intra-peritoneal administration of ABTTA (20 mg/kg) for 4 days in adult WT mice. Analyses were performed 4 days after VEGF-A administration. **(b-c)** Images and comparisons of CD31⁺ vessels in retinas and Dextran (70 kDa) leakage Fc- (Fc, n = 6) or ABTTA- (AB, n = 6) treated WT mice. Error bars represent mean \pm s.d. *P < 0.01, n.s versus Fc by Mann-Whitney U test. All scale bars, 500 μ m (yellow).

Major comment 9. The overall gate-keeping role is centered on endothelial FOXO1, which needs to be kept suppressed to inhibit expression of destabilizing factors, such as Ang2. The authors need at least discuss what are the regulators upstream of FOXO1 and how those FOXO1 inactivators (i.e. PI3K-AKT) affect vascular leakage. In fact, several lines of evidence show that PI3K, as well as Akt inactivation causes lack of maturation and leakage, thereby supporting the authors' conclusions. The authors discuss the manuscript (ref.29), which ironically places FOXO1 upstream of Akt, whereas it is FOXO1 is a direct target of Akt signaling in most scenarios as the authors show in their diagram in Fig.10. This diagram definitely helps to deal with overwhelming amounts of data. Also, the authors have to make a final

connection by discussing how exactly the pericytes keep FOXO1 inactive. There is no direct connection between pericytes and endothelial cells shown on their diagram.

We appreciate this constructive comment.

Regarding the reference (Dharaneeswaran et al., 2014), we cited this article to refer to several FOXO1 downstream target genes, including Ang2. As the reviewer pointed out, according to this study, Akt can be inversely induced by FOXO1 to maintain growth factor responsive Akt/mTORC1 activity within a homeostatic range. However, in this study, the role of Akt was not evaluated in the context of responsiveness to growth factors such as VEGF.

In response to this comment and comment 2, we included additional discussions into the revised text as below (page 12,13).

Mechanistically, the gate-keeping role of intact pericyte-Tie2/Akt signaling, which suppresses FOXO1-mediated transcriptional activation of vascular destabilizing factors including Ang2 in ECs, seems central for the maintenance and protection of BRB integrity against injuries (**Fig. 10d**). The mechanism by which pericyte coverage regulates Tie2 can be explained in two ways. First, Tie2 shedding (cleavage of Tie2 extracellular domain) in ECs could have occurred immediately after pericyte loss, because ECs are directly exposed to a bunch of activated proteases and growth factors such as VEGF-A. In fact, VEGF-A is known to induce Tie2 shedding via a phosphatidylinositol 3 (PI3) kinase/Akt-dependent pathway to modulate Tie2 signaling (Findley et al., 2007). In addition, recent elegant studies (Kim et al., 2016; Korhonen et al., 2016; Singh et al., 2012) have revealed that inflammation-induced cleavage of the Tie1 extracellular domain leads to reduced availability and signaling of Tie2, which in turn further degrades vascular stability through FOXO1-mediated up-regulation of Ang2 in the ECs of tracheal vessels. Thus, in agreement with these studies, inflammation induced by the BRB disruption after pericyte loss could also reduce functional Tie2 level and stimulate FOXO1 activation. Nevertheless, unlike FOXO1 and Ang2, the change in Tie2 upon pericyte depletion was different between growing retinal vessels and stable adult retinal vessels, implying that regulation of Tie2 by pericytes could also be dependent on vessel stability as well as its microenvironment, such as inflammation and hypoxia. PI3 kinase/Akt intracellular signaling is the main downstream pathway for Tie2 activation in ECs (Augustin et al., 2009; Koh, 2013). While PI3 kinases play key roles in the maintenance of vascular integrity and barrier function (Yoshioka et al., 2012; Yuan et al., 2008), Akt also plays a vital role in maintaining vascular integrity against VEGF-induced impairments and increased permeability in blood vessels (Chen et al., 2005; Gao et al., 2016; Kerr et al., 2016). In fact, Akt acts as a negative regulator of FOXO1 by promoting its phosphorylation and degradation (Augustin et al., 2009; Daly et al., 2004). In addition, a previous study has shown that FOXO1 activity is induced together with Akt inactivation in primary cultured ECs of microvasculature under diabetic conditions (Behl et al., 2009). Thus, pericyte loss-induced FOXO1 activation could be due to the reduced Tie2-mediated PI3 kinase/Akt signaling.

(Minor comments)

Minor point 1. In fig 1g. the authors show increased macrophage infiltration in *Pdgfb*^{ΔEC} with F4/80. A significant percentage of these cells could be activated microglia as a consequence of vascular leakage (since F4/80 is used as activated microglial marker). Are the macrophages and microglia observed in the deeper regions of retina even in absence or diminished vascular plexus layer in *Pdgfb*^{ΔEC}?

We appreciate this constructive comment. As noted by the reviewer, a significant portion of the F4/80⁺ cells that heavily infiltrated the retinas of *Pdgfb*^{ΔEC} mice could be activated microglia. As shown in the original Fig. 2g, in response to severe vascular leakage, F4/80⁺ activated microglia invaded the entire retinal layer, including the deep layer, despite the absence of deep vascular plexus in *Pdgfb*^{ΔEC} mice.

Minor point 2. Fig3d: authors show increased Ang2 and ESM1 in *Pdgfb*^{ΔEC}. This could be due to the higher number of ECs as observed with higher CD31 staining in the thicker vessels of *Pdgfb*^{ΔEC}. It might help to demonstrate this point with high magnification images showing individual ECs for comparison of Ang2 levels between WT and *Pdgfb*^{ΔEC}.

To further validate our findings, we additionally visualized Ang2 protein in the retinal vessels of *Pdgfb*^{ΔEC} mice with a higher magnification, and included the images as Supplementary Fig. 3b into the revised manuscript. Strong Ang2 signals with several dots clustered within the retinal vascular ECs, which are similar findings of the previous study (Kim et al., 2016), were observed in *Pdgfb*^{ΔEC} mice, but not noticeable in WT mice.

Minor point 3. Was the Angpt1-GFP- knock-in protein levels comparable to WT?

We appreciate this constructive comment. We tried to compare the protein level by Western blot, but unfortunately there was no antibody that can specifically detect murine Ang1. There are some commercial antibodies, but analyses with these were unreliable and unconvincing. Therefore, we were not able to directly compare the protein level of Ang1 between *Ang1*-GFP-knock-in heterozygote mice and WT mice. Instead, we compared the phenotypes of WT (+/+) mice, *Ang1*-GFP knock-in heterozygote (GFP/+) mice, and *Ang1*-GFP knock-in homozygote (GFP/GFP) mice during embryonic development. We confirmed that embryonic growth and vascular phenotype of embryonic heart between GFP/+ embryos and +/+ embryos were not significantly different. However, GFP/GFP embryo exhibited embryonic lethality and heart defect, which are similar to the findings of conventional Ang1 knock-out embryos (Suri et al., 1996). With these findings, we could assume that the number of allele in *Ang1*-GFP-knock-in mice faithfully represents the Ang1 protein level. We are showing this result only for the reviewer.

The embryos of WT (+/+), *Ang1*-GFP knock-in heterozygote (GFP/+), and homozygote (GFP/GFP) were harvested at embryonic day (E) 10.5 and embryonic hearts were visualized. CD31 staining of vascular endothelium in the developing heart at E10.5 of GFP/GFP embryo exhibited a reduction in size and endocardial folding (arrowheads) compared with those of +/+ and GFP/+ embryos. Scale bars, 500 μ m.

Minor point 4. Can the authors show *Tie2* KO at P10/12 for consistency? What is the phenotype at P30?

The phenotypes of *Tie2*^{iAEC} mice at P10/12 were similar to those at P15 but milder. Therefore, we believed it was better to present the phenotypes at P15. We additionally included the data as Supplementary Fig. 9 and their descriptions (page 8) regarding the phenotype at P30 of endothelial *Tie2* deletion after birth.

In comparison, when *Tie2* was specifically depleted in the ECs by crossing *Tie2*^{flx/flx} mice with *VE-Cadherin-Cre-ER*^{T2} mice (*Tie2*^{iAEC} mice) (**Fig. 5a**), the retinal vessels showed moderate hemorrhage in the inner surface of the retinal cup, severely disrupted vascular network formation, enlarged diameter but short radial length, less density and branching, lack of deep vascular plexus, and profound RBC leakage and macrophage infiltration without a significant difference in pericyte coverage at P15 compared with those of WT mice (**Fig. 5b-g**). No normalization was observed until P30 in the retinas of *Tie2*^{iAEC} mice, which showed vascular leakage with engorged and disorganized vessels (**Supplementary Fig. 9**).

Minor point 5. FOXO1 staining in Fig. 3f appears to be localized to the center of the cell, not just the nucleus. Higher magnification images will be more convincing.

We added more highly magnified and convincing images showing nuclear-localized FOXO1 in the CD31⁺ retinal vessels as Supplementary Fig. 3c in the revised manuscript.

Minor point 6. The rescue with FOXO1 looks extremely convincing. However, the rescue with *Ang2* depletion is much milder. Is it possible FOXO1 regulates transcription of other factors that are essential for vascular stability?

As the reviewer indicated, the rescue against vascular leakage by FOXO1 deletion was stronger than that by Ang2 deletion or Ang2 blockade. As a transcriptional factor, FOXO1 up-regulates the expressions of several key genes including ESM1, VEGFR2, CXCR4, and apelin (Daly et al., 2004; Potente et al., 2005) in addition to Ang2, which are related to vascular destabilization and leakage. To address this comment, we modified the sentences in the revised text (page 9) as follows:

Although it was significant, genetic deletion of Ang2 or Ang2 blockade showed milder effects on relieving the BRB impairments compared with genetic deletion of FOXO1 (Fig. 6,7). Thus, enhanced FOXO1 transcriptional activity upon inadequate pericyte attachment seems to be substantially responsible for deteriorating BRB maturation by up-regulating several vascular destabilization factors including Ang2.

Minor point 7. Why was the rescue with Ang2 blocking Ab shown on P8? How did the vessel diameter compare after the rescue?

We appreciate this constructive comment. We believe that the up-regulated Ang2 induced by destabilization of retinal vessels upon APB5 treatment caused the vascular leakage and micro-aneurysm formation. That was why Ang2 blocking Ab or genetic deletion of Ang2 was able to rescue such pathologic phenotypes. In addition, the antibody treatment was able to reduce vessel diameter. We included this quantification result and its description into the revised manuscript (page 9, Fig. 7d-f).

Moreover, administration of Ang2-blocking antibody also markedly attenuated micro-aneurysm formation, vascular enlargement, and RBC leakage upon APB5 administration (Fig. 7d-f).

Minor point 8. Can developmental abnormalities in $Pdgfb^{\Delta EC}$ or APB5 treated mice be rescued by overexpression of Tie2 or by ABTAA (P12). This might directly inhibit FOXO1 localization to nuclei and decrease Ang2 expression.

Yes, our unpublished results (below) show that the abnormalities, such as vascular leakage and macrophage infiltration, are ameliorated by ABTAA. As the reviewer assumed, it appears to be partly mediated through the inhibition of FOXO1-induced Ang2 expression in the retinal ECs. However, we did not include the results into the original manuscript because we felt that it was repetitive and redundant, because we have presented similar results and their underlying mechanism using the pericyte-deficient adult mouse model ($DTA^{\Delta PC}$ mice).

(a) Diagram depicting the experiment schedule for EC-specific depletion of PDGF-B in retinal vessels from P5, daily i.p ABTAA treatment from P6 to P9, and their analyses at P12 in WT and *Pdgfb*^{ΔEC} mice. (b-d) Images of retinal cup, CD31⁺ retinal vessel, F4/80⁺ macrophage infiltration, and quantification of the indicated parameters in Fc treated (Fc, n = 6) and ABTAA treated (AB, n = 6) mice.

Minor point 9. Fig 7d typo in legend: The description for Fig 7d in the legend labeled as (e).

We corrected this typo error in the revised manuscript.

***References for responses to comments of reviewer 2**

Armulik, A., Genove, G., Mae, M., Nisancioglu, M.H., Wallgard, E., Niaudet, C., He, L., Norlin, J., Lindblom, P., Strittmatter, K., et al. (2010). Pericytes regulate the blood-brain barrier. *Nature* 468, 557-561.

Augustin, H.G., Koh, G.Y., Thurston, G., and Alitalo, K. (2009). Control of vascular morphogenesis and homeostasis through the angiopoietin-Tie system. *Nat Rev Mol Cell Biol* 10, 165-177.

Baker, M., Robinson, S.D., Lechertier, T., Barber, P.R., Tavora, B., D'Amico, G., Jones, D.T., Vojnovic, B., and Hodivala-Dilke, K. (2011). Use of the mouse aortic ring assay to study

angiogenesis. *Nat Protoc* 7, 89-104.

Behl, Y., Krothapalli, P., Desta, T., Roy, S., and Graves, D.T. (2009). FOXO1 plays an important role in enhanced microvascular cell apoptosis and microvascular cell loss in type 1 and type 2 diabetic rats. *Diabetes* 58, 917-925.

Chen, J., Somanath, P.R., Razorenova, O., Chen, W.S., Hay, N., Bornstein, P., and Byzova, T.V. (2005). Akt1 regulates pathological angiogenesis, vascular maturation and permeability in vivo. *Nat Med* 11, 1188-1196.

Daly, C., Wong, V., Burova, E., Wei, Y., Zabski, S., Griffiths, J., Lai, K.M., Lin, H.C., Ioffe, E., Yancopoulos, G.D., et al. (2004). Angiopoietin-1 modulates endothelial cell function and gene expression via the transcription factor FKHR (FOXO1). *Genes Dev* 18, 1060-1071.

Dharaneeswaran, H., Abid, M.R., Yuan, L., Dupuis, D., Beeler, D., Spokes, K.C., Janes, L., Sciuto, T., Kang, P.M., Jaminet, S.C., et al. (2014). FOXO1-mediated activation of Akt plays a critical role in vascular homeostasis. *Circ Res* 115, 238-251.

Findley, C.M., Cudmore, M.J., Ahmed, A., and Kontos, C.D. (2007). VEGF induces Tie2 shedding via a phosphoinositide 3-kinase/Akt dependent pathway to modulate Tie2 signaling. *Arterioscler Thromb Vasc Biol* 27, 2619-2626.

Gao, F., Artham, S., Sabbineni, H., Al-Azayzih, A., Peng, X.D., Hay, N., Adams, R.H., Byzova, T.V., and Somanath, P.R. (2016). Akt1 promotes stimuli-induced endothelial-barrier protection through FoxO-mediated tight-junction protein turnover. *Cell Mol Life Sci* 73, 3917-3933.

Han, S., Lee, S.J., Kim, K.E., Lee, H.S., Oh, N., Park, I., Ko, E., Oh, S.J., Lee, Y.S., Kim, D., et al. (2016). Amelioration of sepsis by TIE2 activation-induced vascular protection. *Sci Transl Med* 8, 335ra355.

Kerr, B.A., West, X.Z., Kim, Y.W., Zhao, Y., Tischenko, M., Cull, R.M., Phares, T.W., Peng, X.D., Bernier-Latmani, J., Petrova, T.V., et al. (2016). Stability and function of adult vasculature is sustained by Akt/Jagged1 signalling axis in endothelium. *Nat Commun* 7, 10960.

Kim, K.T., Choi, H.H., Steinmetz, M.O., Maco, B., Kammerer, R.A., Ahn, S.Y., Kim, H.Z., Lee, G.M., and Koh, G.Y. (2005). Oligomerization and multimerization are critical for angiopoietin-1 to bind and phosphorylate Tie2. *J Biol Chem* 280, 20126-20131.

Kim, M., Allen, B., Korhonen, E.A., Nitschke, M., Yang, H.W., Baluk, P., Saharinen, P., Alitalo, K., Daly, C., Thurston, G., et al. (2016). Opposing actions of angiopoietin-2 on Tie2 signaling and FOXO1 activation. *J Clin Invest* 126, 3511-3525.

Koh, G.Y. (2013). Orchestral actions of angiopoietin-1 in vascular regeneration. *Trends Mol Med* 19, 31-39.

Korhonen, E.A., Lampinen, A., Giri, H., Anisimov, A., Kim, M., Allen, B., Fang, S., D'Amico, G., Sipila, T.J., Lohela, M., et al. (2016). Tie1 controls angiopoietin function in vascular remodeling and inflammation. *J Clin Invest* 126, 3495-3510.

Potente, M., Urbich, C., Sasaki, K., Hofmann, W.K., Heeschen, C., Aicher, A., Kollipara, R., DePinho, R.A., Zeiher, A.M., and Dimmeler, S. (2005). Involvement of Foxo transcription factors in angiogenesis and postnatal neovascularization. *J Clin Invest* 115, 2382-2392.

Savant, S., La Porta, S., Budnik, A., Busch, K., Hu, J., Tisch, N., Korn, C., Valls, A.F., Benest, A.V., Terhardt, D., et al. (2015). The Orphan Receptor Tie1 Controls Angiogenesis and

Vascular Remodeling by Differentially Regulating Tie2 in Tip and Stalk Cells. *Cell Rep* 12, 1761-1773.

Singh, H., Hansen, T.M., Patel, N., and Brindle, N.P. (2012). The molecular balance between receptor tyrosine kinases Tie1 and Tie2 is dynamically controlled by VEGF and TNFalpha and regulates angiopoietin signalling. *PLoS One* 7, e29319.

Suri, C., Jones, P.F., Patan, S., Bartunkova, S., Maisonpierre, P.C., Davis, S., Sato, T.N., and Yancopoulos, G.D. (1996). Requisite role of angiopoietin-1, a ligand for the TIE2 receptor, during embryonic angiogenesis. *Cell* 87, 1171-1180.

Yoshioka, K., Yoshida, K., Cui, H., Wakayama, T., Takuwa, N., Okamoto, Y., Du, W., Qi, X., Asanuma, K., Sugihara, K., et al. (2012). Endothelial PI3K-C2alpha, a class II PI3K, has an essential role in angiogenesis and vascular barrier function. *Nat Med* 18, 1560-1569

Yuan, T.L., Choi, H.S., Matsui, A., Benes, C., Lifshits, E., Luo, J., Frangioni, J.V., and Cantley, L.C. (2008). Class 1A PI3K regulates vessel integrity during development and tumorigenesis. *Proc Natl Acad Sci U S A* 105, 9739-9744.

Reviewer #3 (expert in diabetic retinopathy)**General comments**

Throughout this manuscript, the authors frame the context of their data and draw conclusions which seem story- and not data driven. A series of examples is given below:

Comment 1: The abstract states: ‘Impaired pericyte recruitment to the vessels showed multiple vascular hallmarks of diabetic retinopathy (DR) from BRB disruption.’ When PDGF-B was specifically depleted in ECs of *Pdgfb*^{ΔEC} mice from P5, the mice showed a severe phenotype. Based on the description of the authors, this phenotype is most consistent with widespread capillary insufficiency causing a severe ischemic retinopathy with all its known sequelae, including BRB loss, inflammation, intra-retinal hemorrhage etc. These experiments show that depletion of pericytes by interfering with the PDGF-B/PDGFRβ axis causes a phenotype with similarities to diabetic retinopathy, observations which have been reported in the literature before. In the present study, the authors have not attempted to dissect the sequence of events or causal interrelations of the various aspects of this phenotype. Therefore, their conclusion that BRB disruption has a specific causal role in this phenotype is not supported by any observation or additional experiment and is unwarranted.

In this study, we focused on identifying the roles of pericytes in the BRB and clarifying their molecular mechanisms as summarized in Fig. 10d. Pericyte drop-out by BRB disruption is one of the hallmarks of DR, and we investigated roles of pericytes not only in growing, postnatal retinal vessels and but also in stable, adult retinal vessels. This paper contains several novel findings and the molecular mechanisms related to the roles of pericytes in retinal vessels.

Comment 2: Page 4 states: ‘These data indicate that the impaired BRB maturation caused by the initial poor pericyte coverage is permanent and could continuously worsen neuronal function in the retina, eventually leading to blindness.’ Again, it was not shown by the authors in any way that ‘impaired BRB maturation’ is specifically caused by pericyte depletion, nor was it shown that the deficient BRB in the severely diseased retina at P30 has a specific causal role in this phenotype. These experiments only show that the retina in these experiments was beyond repair after induction of the severe phenotype at P12 and persisting PDGF depletion.

As the reviewer pointed out, the wording ‘maturation’ may have been inappropriate because we did not provide the data related to the maturation process of BRB. However, impaired BRB function caused by initial poor pericyte coverage around P5-12 can be continuously worsen, which led to impaired retinal perfusion and neuronal

damage.

Therefore, we rephrased the sentence (page 6) in revised manuscript as follows:

These data indicate that **impaired BRB function** caused by the initial poor pericyte coverage is permanent **and beyond self-repair**, and could continuously worsen neuronal function in the retina, eventually leading to blindness.

Comment 3: Page 6 states: ‘Thus, adequate pericyte coverage is required to strengthen BRB integrity against further vascular impairments, including severe ischemia, inflammation, and oxidative stresses. The authors have indeed shown that the OIR phenotype is worse in the context of PDGF depletion, but evidence that ‘strengthening the BRB’ would have anything to do with that effect was not provided.

This is valid point that needs to be addressed. In respond to this point, we rephrased the sentence (page 6) in the revised manuscript as follows:

Thus, adequate pericyte coverage is required **to prevent BRB impairment** against further vascular impairments, including severe ischemia, inflammation, and oxidative stresses.

Comment 4: Page 7 states: ‘Thus, the ECs became unstable with inadequate pericyte coverage, and cleavage of Tie1 led to suppression of Tie2 availability for inhibition of FOXO1 activity of Ang2 transcription (32,33). In fact, Chip-PCR sequencing analysis revealed that FOXO1 binding was enriched in promoter and enhancer regions of Ang2 (Supplementary Fig. 3d,e), indicating that FOXO1 directly regulates Ang2 expression in the destabilized ECs caused by inadequate pericyte coverage.’

Although the data presented in this paragraph are interesting and hypothesis-generating, they do not support these conclusions. In particular, the direct role of pericyte coverage in regulating Ang2 expression was not shown. In fact, HUVEC in vitro have no pericyte coverage either but were used as a model anyway.

We agree that this is an important point that needs to be addressed further, more so as the reviewer 2 stressed this point several times throughout several comments. In fact, we already showed a possible connection of inadequate pericyte coverage (by treatment of PDGFR β blocking antibody, APB5)-FOXO1 up-regulation and nuclear localization-Ang2 up-regulation in the ECs as original Fig. 6. We also validated this possible connection using *Foxo1*^{iAEC} in original Fig. 6.

Nevertheless, to further address this connection, we performed an additional experiment (Supplementary Fig. 5) by employing an *ex vivo* aortic ring assay (Baker et al., 2011) using the aortas of WT and *Pdgfb*^{iAEC} mice. With this assay we were able to detach PDGFR β ⁺ or NG2⁺ pericytes from ECs in sprouting vessels of the aorta of *Pdgfb*^{iAEC} mice, while keeping the PC-EC interaction intact in those of the

aorta from WT mice. Moreover, this assay excludes the possible effects from neurohormonal cells or inflammatory cells, which occur *in vivo*. The pericyte-detached ECs exhibited high levels and nuclear localization of FOXO1 and high level of Ang2, and intriguingly the detached pericytes also exhibited high levels and nuclear localization of FOXO1 (Supplementary Fig. 5). In contrast, no major changes in FOXO1 and Ang2 were detected in the sprouting vessels with intact pericytes in the aorta of WT mice (Supplementary Fig. 5).

Furthermore, we performed an additional experiment with DTA^{ΔPC} mice to investigate this connection (Supplementary Fig. 15). Pericyte detachment in the growing retinal vessels not only up-regulated FOXO1 but also induced nuclear localization of FOXO1 as well as up-regulating Ang2 in the ECs. Thus, pericyte detachment appears to up-regulate and activate FOXO1 directly in ECs of the retinal vessels, which leads to stimulation of Ang2 expression in ECs of the vessels.

We included these results and related description into the revised manuscript (page 7, 10)

Next, to reveal whether and how pericyte coverage regulates FOXO1 and Ang2, we employed an *ex vivo* aortic ring assay (Baker et al., 2011) using the aortas of WT and *Pdgfb*^{ΔEC} mice, where the mice were pre-administrated with tamoxifen before aorta harvest (Supplementary Fig.5a). During *ex vivo* culture, the PDGFRβ⁺ pericytes were detached from the sprouts, which were reduced in number and length themselves, of the aorta of *Pdgfb*^{ΔEC} mice, and the pericyte-detached ECs had higher levels of Ang2 and FOXO1 as well as nuclear localization of FOXO1 (Supplementary Fig. 5b-e). In contrast, no major changes in FOXO1 or Ang2 were detected in the ECs of sprouts with intact pericytes in the aorta of WT mice (Supplementary Fig. 5d,e). Thus, pericyte detachment appears to directly up-regulate and activate FOXO1 in ECs, which leads to stimulation of Ang2 expression in ECs. In fact, Chip-PCR sequencing analysis revealed that FOXO1 binding was enriched in the promoter and enhancer regions of Ang2 (Supplementary Fig. 6a,b), indicating that FOXO1 directly regulates Ang2 expression in the destabilized ECs caused by inadequate pericyte coverage (page 7).

On the other hand, impaired retinal vascular maturation together with BRB disruption, which were similar to the findings of *Pdgfb*^{ΔEC} mice, were observed in DTA^{ΔPC} mice during postnatal development (Supplementary Fig. 15a-c). These findings indicate that the regulation of pericytes on BRB could be dependent on stability of the vessels, and a sudden pericyte loss from the stabilized retinal vessels is insufficient to induce a significant alteration in the integrity of retinal vessels and BRB, and additional insults may be required to mimic the hallmarks of DR (page 10).

Comment 5: The abstract states: ‘Accordingly, either blocking Ang2 or activating Tie2 greatly ameliorated progression of BRB breakdown, providing beneficial effects in delaying DR progression by preventing BRB disintegration.

In the adult mice with reduced pericyte coverage caused by DTA^{ΔPC} mice, the

BRB was not affected, arguing against a major maintenance role of pericytes in the BRB in mice. The authors show data that this led to increased BRB loss from a single injection of exogenous VEGF-A and a retinopathy similar to the previously reported effects of repeated exogenous VEGF-A injections in monkeys. The observed effects of VEGF-A in mice in this study could be blocked by Tie2 activation. These observations are all very interesting and mostly novel. However, the effect of Ang2 inhibition mentioned in the abstract in this context cannot be retrieved in the Results section.

Furthermore, what is meant by 'progression of BRB breakdown', as the experiment was based on one time point only?

More importantly, the extrapolation to DR progression by the authors and the claim that DR progression could be delayed by preventing BRB disintegration is farfetched. DR progression in mice is not identical to increasing BRB loss, but is characterized by vasoregression, the loss of capillaries.

First, as our group previously reported, the action of ABTAA is attributed to both Ang2 inhibition and Tie2 activation (Han et al., 2016), which is why we mentioned 'blocking Ang2' in the sentence of the abstract the reviewer pointed out.

Second, as the reviewer pointed out, DR progresses as a result capillaries loss and subsequent development of non-perfused ischemic retina, which produce abundant VEGF-A. From this background, we thought that the finding of BRB impairment upon VEGF-A treatment in pericyte-depleted retinal vessels is similar with the consequences of DR progression. Considering this, we modified two sentences in our revised manuscript as follows (page 2, 4, 11):

Accordingly, either blocking Ang2 or activating Tie2 greatly ameliorated ~~progression of~~ BRB breakdown, providing potentially beneficial effects on retinal damages upon DR progression (page 2).

This process was inhibited by Ang2 blockade or Tie2 activation, providing beneficial effects on retinal damages during DR progression by inhibiting BRB disintegration (page 4).

Thus, Tie2 activation could ameliorate BRB disintegration against VEGF-A stimuli in the absence of pericytes, preventing retinal damages during DR progression (page 11).

*References for responses to comments of reviewer 3

Baker, M., Robinson, S.D., Lechertier, T., Barber, P.R., Tavora, B., D'Amico, G., Jones, D.T., Vojnovic, B., and Hodivala-Dilke, K. (2011). Use of the mouse aortic ring assay to study angiogenesis. Nat Protoc 7, 89-104.

Han, S., Lee, S.J., Kim, K.E., Lee, H.S., Oh, N., Park, I., Ko, E., Oh, S.J., Lee, Y.S., Kim, D., et al. (2016). Amelioration of sepsis by TIE2 activation-induced vascular protection. Sci Transl Med 8, 335ra355.

Reviewer #1 (Remarks to the Author):

The authors have answered the comments, by providing additional information and new data to explain some of the controversies of the original manuscript. Thus, the manuscript is significantly improved, and there are only few points that should be considered.

1. In their response, the authors point to a differential requirement of the lung and retinal vessels for Tie2 ("...lung vessels continue to require Tie2, while the retinal vessels no longer require Tie2 during adulthood."). Including this information in the manuscript would make the authors' results more understandable with regards to previously published literature.
2. P. 12 Discussion. "In the context of Tie2, we showed that, unlike Ang1, endothelial Tie2 plays an indispensable role in vascular development and BRB maturation." This statement remains unclear, since the authors discuss and have previously reported that global Ang1 deletion is deleterious during retinal vascular development.
3. P. 12 Discussion. Please, add a reference where it has been shown that "... the clustering of Tie2 can also be strongly induced by shear stress, which is an independent mode of action to Ang1-induced activation"

Reviewer #2 (Remarks to the Author):

This is a very strong and well-documented study on the signaling mechanisms underlying the complexity of the pericyte-mediated stabilization of the blood-brain barrier. The manuscript by Koh and colleagues underwent a thorough and constructive revision resulting in a very impressive and elegant story.

The authors carefully addressed all the comments raised during the previous review, incorporated lots of new data and revised the discussion. The amount of the data (especially in vivo data) supporting the conclusions of this manuscript is truly impressive and there is no need to make this paper any longer or any more complex. I think this is a very strong manuscript of high significance.

Answers to the editor's and reviewers' comments

Reviewer #1

The authors have answered the comments, by providing additional information and new data to explain some of the controversies of the original manuscript. Thus, the manuscript is significantly improved, and there are only few points that should be considered.

1. In their response, the authors point to a differential requirement of the lung and retinal vessels for Tie2 (“...lung vessels continue to require Tie2, while the retinal vessels no longer require Tie2 during adulthood.”). Including this information in the manuscript would make the authors' results more understandable with regards to previously published literature.

We appreciate this comment. We added this statement into the revised manuscript (page 8).

However, when Tie2 was specifically depleted in the ECs at adulthood, no noticeable changes including vascular leakage were detected in the retinal vessels compared with those of WT mice (**Supplementary Fig. 10**). These results indicate that Tie2 is indispensable for vascular growth and BRB maturation in the growing retinal vessels, but is dispensable for the maintenance of BRB integrity during adulthood. *In fact, lung vessels continue to require Tie2 (Frye et al., 2015), while retinal vessels no longer require Tie2 during adulthood, suggesting that dependency on Tie2 for vascular stability and integration could be organ-specific during adulthood.*

2. P. 12 Discussion. “In the context of Tie2, we showed that, unlike Ang1, endothelial Tie2 plays an indispensable role in vascular development and BRB maturation.” This statement remains unclear, since the authors discuss and have previously reported that global Ang1 deletion is deleterious during retinal vascular development.

We appreciate this critical comment. We revised this statement as follows.

In the context of Tie2, we showed that, ~~unlike Ang1~~, endothelial Tie2 plays an indispensable role in vascular development and BRB maturation.

3. P. 12 Discussion. Please, add a reference where it has been shown that "... the clustering of Tie2 can also be strongly induced by shear stress, which is an independent mode of action to Ang1-induced activation"

We appreciate this comment. We modified the sentences with additional reference (Lee and Koh 2003), and we rearranged this paragraph as follows with regard to the

answer to comment 2.

In the context of Tie2, we showed that endothelial Tie2 plays an indispensable role in vascular development and BRB maturation. It is well-known that Ang1-induced Tie2 clustering is critical for its activation (Augustin et al., 2009; Kim et al., 2005; Koh, 2013). However, the activation of Tie2 can also be strongly induced by shear stress (Lee and Koh, 2003), which is an independent mode of action to Ang1-induced activation. These collectively imply that Ang1 may have derived from adjacent neurons or hematopoietic cells including platelets, which subsequently activate Tie2 or integrins for its roles in formation and maturation of the inner part of the BRB, or the abnormal vascular phenotypes after endothelial Tie2 deletion can be due to the absence of Tie2 activation induced by a blood flow-mediated shear stress.

Reviewer #2

This is a very strong and well-documented study on the signaling mechanisms underlying the complexity of the pericyte-mediated stabilization of the blood-brain barrier. The manuscript by Koh and colleagues underwent a thorough and constructive revision resulting in a very impressive and elegant story. The authors carefully addressed all the comments raised during the previous review, incorporated lots of new data and revised the discussion. The amount of the data (especially in vivo data) supporting the conclusions of this manuscript is truly impressive and there is no need to make this paper any longer or any more complex. I think this is a very strong manuscript of high significance.

We really appreciate these favorable and encouraging comments.

***References for responses to comments of reviewers**

Augustin, H.G., Koh, G.Y., Thurston, G., and Alitalo, K. (2009). Control of vascular morphogenesis and homeostasis through the angiopoietin-Tie system. *Nat Rev Mol Cell Biol* 10, 165-177.

Frye, M., Dierkes, M., Kuppens, V., Vockel, M., Tomm, J., Zeuschner, D., Rossaint, J., Zarbock, A., Koh, G.Y., Peters, K., *et al.* (2015). Interfering with VE-PTP stabilizes endothelial junctions in vivo via Tie-2 in the absence of VE-cadherin. *The Journal of experimental medicine* 212, 2267-2287.

Kim, K.T., Choi, H.H., Steinmetz, M.O., Maco, B., Kammerer, R.A., Ahn, S.Y., Kim, H.Z., Lee, G.M., and Koh, G.Y. (2005). Oligomerization and multimerization are critical for angiopoietin-1 to bind and phosphorylate Tie2. *J Biol Chem* 280, 20126-20131.

Koh, G.Y. (2013). Orchestral actions of angiopoietin-1 in vascular regeneration. *Trends Mol*

Med 19, 31-39.

Lee, H.J., and Koh, G.Y. (2003). Shear stress activates Tie2 receptor tyrosine kinase in human endothelial cells. *Biochem Biophys Res Commun* 304, 399-404.